# Convergence of plasmid-driven virulence and antibiotic resistance in *Escherichia coli*

Zheng Jie Lian [1,2], Nguyen Thi Khanh Nhu [1,2], Chitra Ravi[1,2,4], Chyden Chang [1,2], Irene Martinez-Roman[1,2], Minh-Duy Phan [1,2] ✉ & Mark A. Schembri [1,2,3] ✉

Plasmids are major vehicles for the spread of antibiotic resistance genes. Some plasmids additionally carry virulence genes that enhance host pathogenicity. The convergence of resistance and virulence genes on the same plasmid poses significant risk, providing a mechanism to create pathogens that cause severe disease with limited treatment options. Colicin V (ColV)-like plasmids (ColVLPs) are virulence plasmids frequently carried by extra-intestinal pathogenic *E. coli* (ExPEC) that cause human and avian infection. Here, by generating and analysing a ColVLP database, we demonstrate that ColVLPs form four distinct sub-groups, characterised by genes encoding for Colicins V and M, with differing virulence and antimicrobial resistance gene carriage. Three of these sub-groups possess moderate-high resistance towards multiple antibiotic classes. We further describe ColVLP co-integrates that have acquired extensive resistance profiles, including against last line colistin, through recombination with co-resident plasmids. Using pMS7163A, a ColVLP from a virulent ExPEC strain, we also demonstrate that the ColVLP-encoded outer membrane protease virulence factor OmpTp works co-operatively with its chromosomal homolog to enhance ExPEC resistance against human cathelicidin (LL-37), an antimicrobial peptide expressed in the urinary tract. Together, our work characterises ColVLPs as high-risk mobile genetic elements that amplify the convergence of resistance and virulence in ExPEC.

Antibiotic resistance is a critical threat to global human health. Plasmids are a major driver contributing to this problem as they facilitate the horizontal dissemination of resistance genes via conjugation[1]. However, some plasmids are known to carry cargo genes encoding potent virulence factors[2]. The combined carriage of virulence and resistance determinants on the same plasmid has the potential to enhance the capacity of host bacteria to cause infections with reduced treatment options.

Colicin V (ColV) plasmids are a subset of Incompatibility (Inc) group F plasmids associated with enhanced virulence of extra-intestinal pathogenic *Escherichia coli* (ExPEC) that cause disease in animals and humans[3,4]. These plasmids are most frequently found in avian pathogenic *E. coli* (APEC) that cause infection in poultry[5], but are also found in several ExPEC clones that cause human infections, including uropathogenic *E. coli* clones that cause urinary tract infection (UTI) and sepsis, such as ST131[6], ST95[7,8] and ST58[9]. ColV is a peptide antibiotic (microcin) that can kill or inhibit the growth of some *E. coli* and other closely related bacteria[10]. ColV production involves genes encoding its biosynthesis (*cvaABC*) and protection is conferred by an immunity protein (*cvi*) that prevents self-intoxification[11,12]. Although the *cvaABC* and *cvi* genes are markers of ColV plasmids, ColV production does not contribute to virulence[13]. Rather, increased

[1]Institute for Molecular Bioscience, The University of Queensland, Brisbane, QLD, Australia. [2]Australian Infectious Diseases Research Centre, The University of Queensland, Brisbane, QLD, Australia. [3]School of Chemistry and Molecular Biosciences, The University of Queensland, Brisbane, QLD, Australia. [4]Present address: UQ Centre for Clinical Research, The University of Queensland, Brisbane, QLD, Australia. ✉e-mail: m.phan1@uq.edu.au; m.schembri@uq.edu.au

virulence is associated with conserved cargo genes carried on ColV plasmids that include: *sitABCD* (metal transporter), *shiF* (aerobactin auxiliary factor), *iucABCD/iutA* (aerobactin siderophore), *hlyF* (upregulates outer membrane vesicle production), *ompTp* (involved in resistance to human cationic peptides), *etsABC* (ATP-binding cassette transporter), *iss* (involved in serum resistance) and *iroBCDEN* (salmochelin siderophore)[4,14,15].

Recent studies have identified a group of plasmids closely related to ColV plasmids, termed ColB/M plasmids[16]. ColB/M plasmids carry most of the ColV-associated virulence cargo genes but not ColV itself[6,7,16]. The shared characteristics between ColB/M and ColV virulence plasmids have led to their redefinition collectively as ColV-like plasmids (ColVLPs), with the presence of ColV genes a marker (but not a strict requirement) for identification. Most ColVLP-encoded virulence loci are also found on the ExPEC chromosome within genomic islands, which include *sitABCD, iucABCD/iutA, iss, iroBCDEN, ompT* and *shiF*[17–19]. Prior studies have indicated sequence diversity between the ColVLP and chromosomal homologous genes[4,20], but a comprehensive comparison of ColVLP and chromosomal homologues in terms of their evolutionary trajectory, virulence properties and potential synergistic effects is lacking. ExPEC that carry ColVLPs are associated with infections at the severe end of the UTI spectrum (e.g. acute pyelonephritis) and sepsis[17]. Most importantly, several studies have reported increased antimicrobial resistance (AMR) among *E. coli* strains carrying ColVLPs[9,21], with some studies highlighting the mobile genetic element-mediated insertion of these AMR determinants on the ColVLP plasmid backbone—a concerning convergence of virulence and resistance on a conjugative mobile genetic element[22].

Here, by collecting and analysing a database of completely sequenced ColVLPs, we describe a plasmid clustering profile associated with differences in AMR gene carriage as well as the existence of ColVLP co-integrates that carry resistance to last-line antibiotics. In terms of virulence, we demonstrate that the ColVLP-encoded outer membrane protease virulence factor OmpT (referred to in this study as OmpTp) contributes to ExPEC resistance to human cathelicidin (LL-37), a soluble antimicrobial peptide that protects against UTI[23,24]. Collectively, our work highlights the important roles of ColVLPs in augmenting ExPEC pathogenesis and resistance, an important advance given the high rates of infection caused by ExPEC clones that carry ColVLPs.

## Results

### ColVLPs can be split into six clusters based on gene content
We generated a dataset comprising 233 non-redundant, completely sequenced ColVLPs identified from PLSDB[25] and NCBI RefSeq[26] according to previously described criteria[6]. This stipulated that the ColVLPs contained >1 gene from at least four of the following sets: (i) *cvaABC* and *cvi*; (ii) *iroBCDEN*; (iii) *iucABCD* and *iutA*; (iv) *etsABC*; (v) *ompT* and *hlyF*; (vi) *sitABCD*. Plasmid taxonomic unit (PTU) classification based on average nucleotide identity revealed that most ColVLPs (225/233) belonged to, or were closely related to, PTU-F$_E$, conjugative F plasmids that reside in *E. coli*[27] (Supplementary Fig. 1).

Bacteria carrying these ColVLPs were isolated from multiple sources, with the most common based on available metadata being human (35%; 50/143), avian (29%; 42/143), porcine (18%; 26/143) and environmental (7%; 10/143) isolates (Fig. 1). Most ColVLPs were isolated from *E. coli* (96%; 224/233), and less frequently from *S. enterica* (6/233) and *K. pneumoniae* (3/233). Most ColVLPs contained more than one F replicon, with the most common combinations (using the FII:FIA:FIB formula[28]) being 18:-:1 (38%; 90/233), 24:-:1 (17%; 41/233), 2:-:1 (10%; 25/233) and 18:5,6:1 (5%; 12/233) (Fig. 1). When replicon carriage across the 233 ColVLPs was queried independently, the FIB_1 (85%; 199/233) and FII_18 (50%; 118/233) replicons were dominant. More than half of the ColVLPs (52%) carried all genes required for F plasmid conjugation and

are potentially conjugative (Fig. 1), although transfer region polymorphisms and/or insertions may still impact transfer capability[29].

Clustering of ColVLPs using an ORF-based analysis tool[30,31] revealed two minor clusters (clusters 1 and 2) and four major clusters (clusters 3–6) (Fig. 1). A defining feature of the major clusters was the presence of colicins. Most ColVLPs of clusters 3 and 6 (>70%) encoded a complete ColV operon (encoding both the peptide antibiotic and the immunity protein), while ColVLPs of clusters 4 and 5 encoded a complete ColM operon (>80%) and the ColV and ColB immunity genes (>80%). The major clusters also differed in their virulence gene carriage, with >84% prevalence of all virulence loci (*iroBCDEN, etsABC, ompTp, hlyF, iucABCD/iutA, shiF, sitABCD, iss* and *traT*) in clusters 3 and 6, and lower virulence factor carriage for clusters 4 (*iroBCDEN*−76%; *etsABC*−17%; *iucABCD/iutA*−44%; *shiF*−43%; *iss*−75%; *traT*−81%) and 5 (*etsABC*−19%; *iucABCD/iutA*−69%; *shiF*−69%; *traT*−19%) (Fig. 1). The prevalence of AMR genes between these major clusters also differed. ColVLPs from cluster 3 had low carriage rates of all AMR genes examined, while ColVLPs of cluster 6 had moderate rates of resistance towards aminoglycosides, beta-lactams, sulphonamides, tetracyclines and trimethoprim antibiotics (~35–45%) (Fig. 1). ColVLPs from clusters 4 and 5 possessed high AMR gene carriage across multiple antibiotic classes including aminoglycoside (~65% and ~100%), beta-lactam (~75% and ~75%), fluoroquinolone (~25% and ~45%), macrolide (~25% and ~50%), phenicol (~35% and ~70%), tetracycline (~75% and ~55%) and trimethoprim (~55% and ~90%) antibiotics (Fig. 1). The presence/absence of each AMR gene determinant is detailed in Supplementary Data 4 and visualised in Supplementary Fig. 2, as the independent acquisition of each determinant can inform ColVLP evolutionary history, particularly in the case of rare determinants such as *sul3*[29]. Taken together, the analysis revealed that ColVLPs can be broadly divided into two major groups: (i) clusters 4 and 5, characterised by the carriage of ColM and the association with lower virulence and higher AMR gene carriage, or (ii) clusters 3 and 6, characterised by the carriage of ColV and the association with higher virulence and low-moderate AMR gene carriage.

### ColVLP variation includes the formation of co-integrate plasmids
Outside of the major clusters, our analysis also identified minor clusters 1 and 2, which consisted of three plasmids each that carried additional IncHI2/IncHI2A and IncN replicons, respectively (Supplementary Fig. 3). In addition to ColVLP-associated virulence loci and numerous AMR genes, these hybrid plasmids also carry *mcr-1* (Cluster 1: 2/3; Cluster 2: 3/3; Fig. 1), conferring resistance to the last-line antibiotic colistin. A detailed analysis of these plasmids revealed two distinct lineages of ColVLP hybrids that contain regions from IncHI2-like plasmids (Supplementary Fig. 4), likely acquired from co-integration events. Cluster 1 ColVLPs (isolated from China) had significant sequence homology to the reference ColVLP pMS7163A[32] and the IncHI2 plasmid pSTM6-275[33] across regions encoding for replication, virulence and transfer (Fig. 2A; Supplementary Fig. 4). Cluster 2 ColVLPs (isolated in the USA) similarly had significant sequence homology to the aforementioned reference plasmids. However, these plasmids lacked an IncHI2 replicon and instead carried an IncN replicon, despite no significant homology to the IncN reference plasmid R46 (AY046276.1)[34] (Fig. 2A; Supplementary Fig. 4).

IS*26*, an insertion element associated with the mobilisation of resistance genes[35], has been previously implicated in co-integrate formation[36] and is known to be present in both ColVLPs[22,29,37] and IncHI2-like plasmids[33,38]. Investigation of our co-integrate plasmids for IS*26* revealed that all co-integrates harboured multiple IS*26* copies (between 6-15 copies) in regions between the ColVLP and co-resident plasmid (Fig. 2A), consistent with previous reports that demonstrate the role of IS*26* in co-integrate formation[35,36,39,40].

## ColVLP co-integrates carry high AMR gene loads

The presence of co-integrated ColVLPs prompted a wider search for additional, more recent chimeric ColVLPs outside of our collection. By querying NCBI for ColVLPs using the criteria of Liu et al.[6] and PlasmidFinder replicons[41], we identified 20 non-redundant ColVLPs which contained additional IncHI1, IncHI2, IncI1, IncN, IncX, or IncR replicons (Supplementary Data 1). Pairwise comparisons confirmed regions of high similarity to their respective reference plasmids, except for two IncX plasmids (AP017611.1 and CP0916721.1) (Supplementary Figs. 5–10). Of these 20 plasmids, 12 had available whole genome assembly data, allowing us to perform MLST classification of their host chromosomes. These ColVLP co-integrates were isolated from diverse *E. coli* STs, including ST101, ST141, ST93, ST23, ST457, ST744, ST359, ST1196, ST88 and ST117 (Supplementary Data 1). Importantly, these STs all belong to the ExPEC/APEC pathotype associated with human, avian and/or other animal infections[42–48].

Sequence analysis of these 20 co-integrates revealed five clusters of closely related plasmids (largely corresponding to their co-integrant Incompatibility group): IncHI2+, IncN+, IncX1+, IncHI1+ and IncN/IncR+ (Fig. 2B). The five plasmids in the IncHI2+ cluster (of which two were *mcr-1* positive) were isolated from human, avian, porcine and fish sources, suggestive of zoonotic transmission and/or conjugation between human and animal isolates. All IncN+ ColVLPs were *mcr-1* positive and from human isolates. As these hybrid plasmids were the result of co-integration events, we hypothesised that they would also acquire the AMR repertoire of their co-integrant plasmid. Indeed, a comparison of the number of AMR genes revealed significantly more AMR genes on ColVLP co-integrates compared to their non-hybrid counterparts ($p < 0.0001$; Fig. 2C). Interestingly, all co-integrates except for one IncI1+ plasmid contained IS*26* insertion elements, mostly located in regions between the ColVLP and co-resident plasmid (Supplementary Figs. 5–10), further supporting the role of IS*26* in co-integrate formation. Overall, our identification of these ExPEC- and APEC-associated ColVLP co-integrates identified an underlying group of high-risk plasmids with the potential to confer both increased antibiotic resistance and enhanced virulence to *E. coli* with strong zoonotic potential.

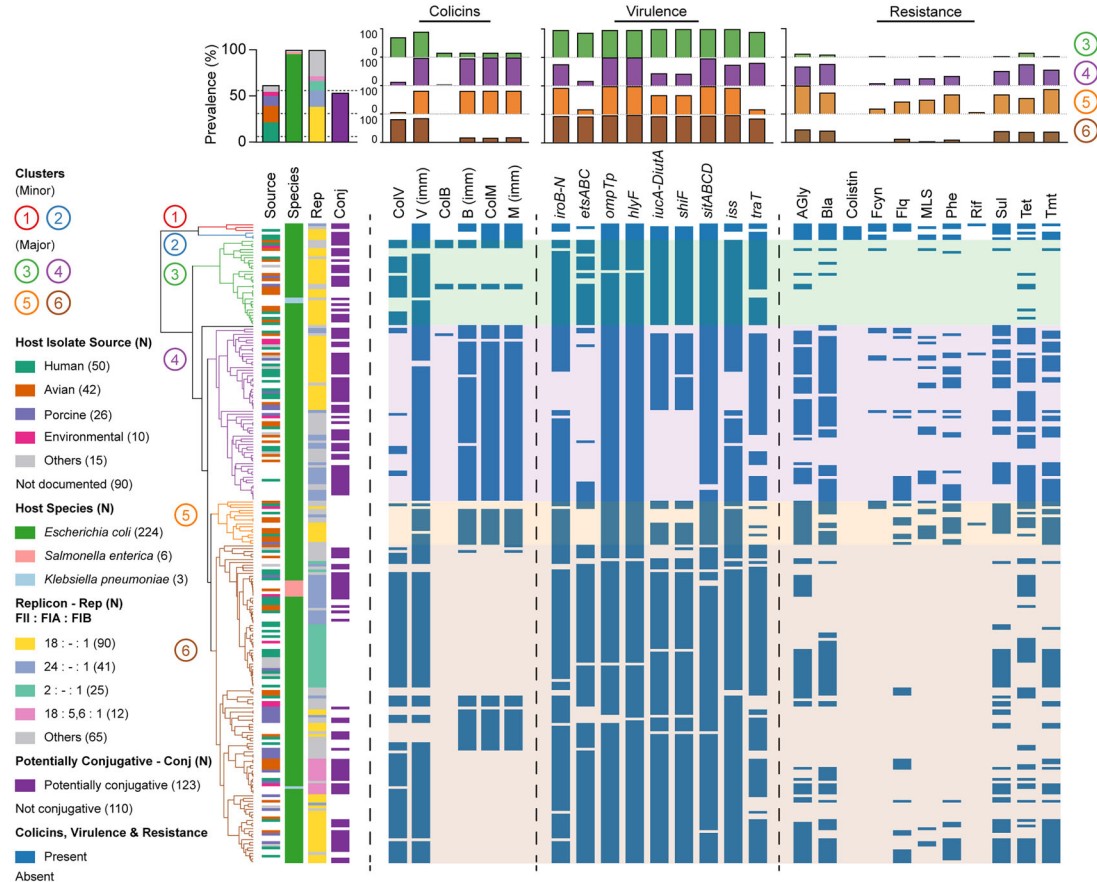

**Fig. 1 | Cladogram of 233 ColV-like plasmids (ColVLPs).** Unique ORFs from 233 ColVLPs were combined to form a hypothetical plasmid. Using sequence similarity searches against the hypothetical plasmid at an 80% nucleotide sequence identity and alignment length threshold, a binary sequence indicating open-reading frame presence/absence for each plasmid was generated. Binary sequences were subjected to hierarchical clustering using Manhattan distance and visualised as a cladogram. Plasmids were arranged into clusters ($n > 1$) based on the total within sum of squares method, with minor ($n \le 5$) and major clusters ($n > 5$). Host isolate source, host species, replicon type, predicted conjugative ability and the presence of colicins, colicin immunity genes (imm), virulence and resistance features are annotated. Feature prevalence is illustrated in a bar plot above the metadata. Potential conjugative ability was predicted based on the presence of all genes required for F plasmid conjugation[82] determined using BLASTn at a 90% nucleotide sequence identity and alignment length threshold. Replicons were identified using ABRicate v0.8 against the PlasmidFinder database at a 100% query length threshold. Virulence genes were identified using a BLASTn search at a 90% nucleotide sequence identity and 95% alignment length threshold, with operons being considered present only if all genes were individually identified. Antimicrobial resistance genes were identified using ABRicate v0.8 against the ARG-ANNOT database[84] using a 100% query length threshold and grouped based on antibiotic class, with the following abbreviations: AGly aminoglycosides, Bla beta-lactamases, Fcyn fosfomycin, Flq fluoroquinolones, MLS macrolide-lincosamide-streptogramin, Phe phenicols, Rif rifampin, Sul sulphonamides, Tet tetracyclines and Tmt trimethoprim. Source data are provided as a Source data file in Supplementary Data 4.

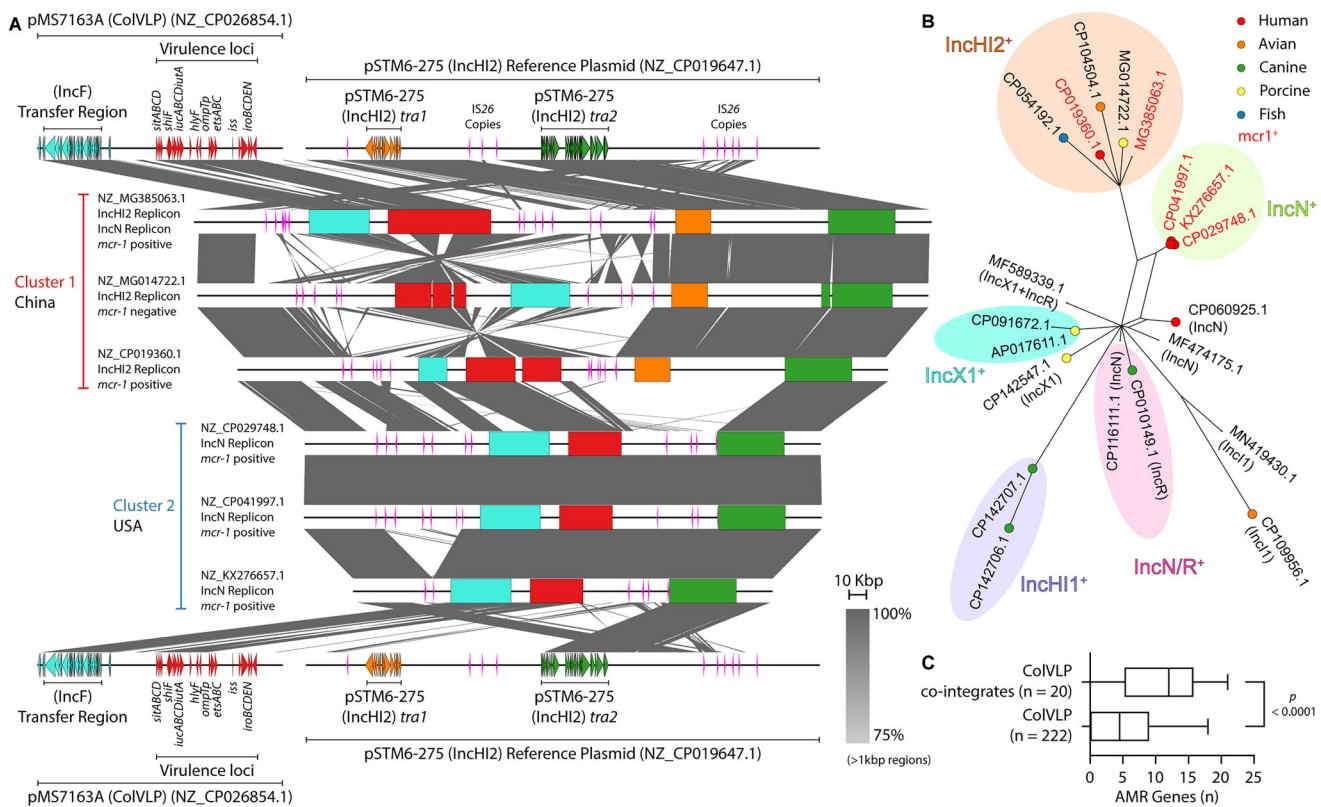

**Fig. 2 | ColV-like plasmid (ColVLP) co-integrates. A** Pairwise comparisons of IncHI2/ColVLP co-integrate plasmids. Reference sequences were a concatenation of the reference ColVLP pMS7136A (NZ_CP026854.1) and the reference IncHI2 plasmid pSTM6-275 (NZ_CP019647.1). Pairwise comparisons were performed using BLASTn and visualised using EasyFig[81] with a minimum alignment length of 1000 bp. Features are coloured as follows: IncF *tra* transfer region–blue; ColVLP virulence loci–red; IncHI2 *tra1* transfer region–orange; IncHI2 *tra2* transfer region –green; IS*26*–pink. **B** Split decomposition network of ColVLP co-integrates. Twenty non-redundant ColVLP co-integrates were identified from NCBI using the criteria of Liu et al.[6] and PlasmidFinder replicons[42]. A network visualising plasmid relatedness was drawn using binary sequences generated using the open-reading-frame-based binarised structure network analysis tool. Clusters were determined based on a Dice index of >0.71. **C** Number of AMR genes (*n*) in ColVLP co-integrates and ColVLPs. Antimicrobial resistance genes were identified using ABRicate v0.8 against the ARG-ANNOT database[84] using a 100% query length threshold. Data is presented as a boxplot, where the box limits represent first and third quartiles, the internal line represents the median, and whiskers represent data within a ±1.5 interquartile range. Significance was determined using an unpaired, two-tailed *t*-test. Figure acronyms: AMR antimicrobial resistance.

## ColVLP/AMR plasmid co-integration is likely driven by ST-specific plasmid carriage patterns

ColVLP co-integrates can arise when ColVLPs recombine with co-resident plasmids. To better understand this phenomenon, we first assessed the frequency at which ColVLPs are carried with additional plasmids by utilising a draft genome database consisting of 9962 *E. coli* genomes from the top 100 STs in Enterobase (100ST database)[49] and querying for the presence of ColVLPs and additional non-IncF replicons, assuming that distinct replicons were located on different plasmid backbones. We found ColVLPs in most STs, with a strong association with human and avian isolates (Supplementary Fig. 11). Across all ColVLP+ strains in the 100ST database (1469/9962; Fig. 3A), we also found that most strains either carried the ColVLP alone (~40%) or carried one additional non-IncF replicon (~40%), although up to five additional non-IncF replicons were found in some strains (Fig. 3A, G). At the ST level, the distribution of additional replicons was ST-specific, with >50% of strains in some STs carrying ColVLPs alone (ST73, ST95, ST117, ST162, ST3580), and other STs with >50% of strains carrying ColVLPs with one or more additional non-IncF replicon (ST23, ST57, ST88, ST428, ST602) (Fig. 3B). Notably, ST131 was absent from the top 20 ColVLP+ STs, as ColVLP carriage is specific to the ST131 Clade B sublineage (55.5%; Supplementary Fig. 12).

To investigate the replicons associated with ColVLP carriage, we plotted all significant co-carriage events in the 100ST database as a network. The top four replicons associated with ColVLP carriage were Incl1, IncX, IncB/O/K/Z and IncH (Fig. 3C). These co-carriage associations were also ST-dependent, as highlighted by the ColVLP-replicon associations of ST95 and ST131 (Clade B), two pandemic STs typically associated with ColVLP carriage, which were quantified with ST-specific databases of 2118 and 344 draft genomes, respectively. In ColVLP+ ST95 (1268/2118; Fig. 3F), the top four replicons associated with ColVLP carriage were (in order of magnitude): IncB/O/K/Z, IncX, IncH and Incl1 (Fig. 3D). In ColVLP+ ST131 Clade B genomes (191/344; Fig. 3F), the top four replicons were: Incl1, IncH, IncX and IncN (Fig. 3E). Notably, strains of ST131 Clade B had higher replicon diversity compared to ST95. ST131 Clade B strains, in addition to carrying ColVLPs alone (31%), frequently contained combinations of ColVLP + Incl1 (24%); + Incl1 and IncH (10%); or + IncX (7%) (Fig. 3I). In contrast, the majority of ST95 carried ColVLP alone (75%) or with an additional IncB/O/K/Z plasmid (13%) (Fig. 3H). Taken together, the data demonstrate that ColVLP+ *E. coli* strains frequently carry a diverse array of plasmid replicons, in part reflecting the existence of multiple co-resident plasmids in ST-specific patterns.

## Virulence genes on ColVLPs are distinct from their chromosomal homologues

Several virulence determinants frequently associated with ColVLPs are also found on the chromosome, including the siderophores salmochelin (*iroBCDEN*) and aerobactin (*iucABCD/iutA*), the metal acquisition system (*sitABCD*), the increased serum survival gene (*iss*), the outer membrane

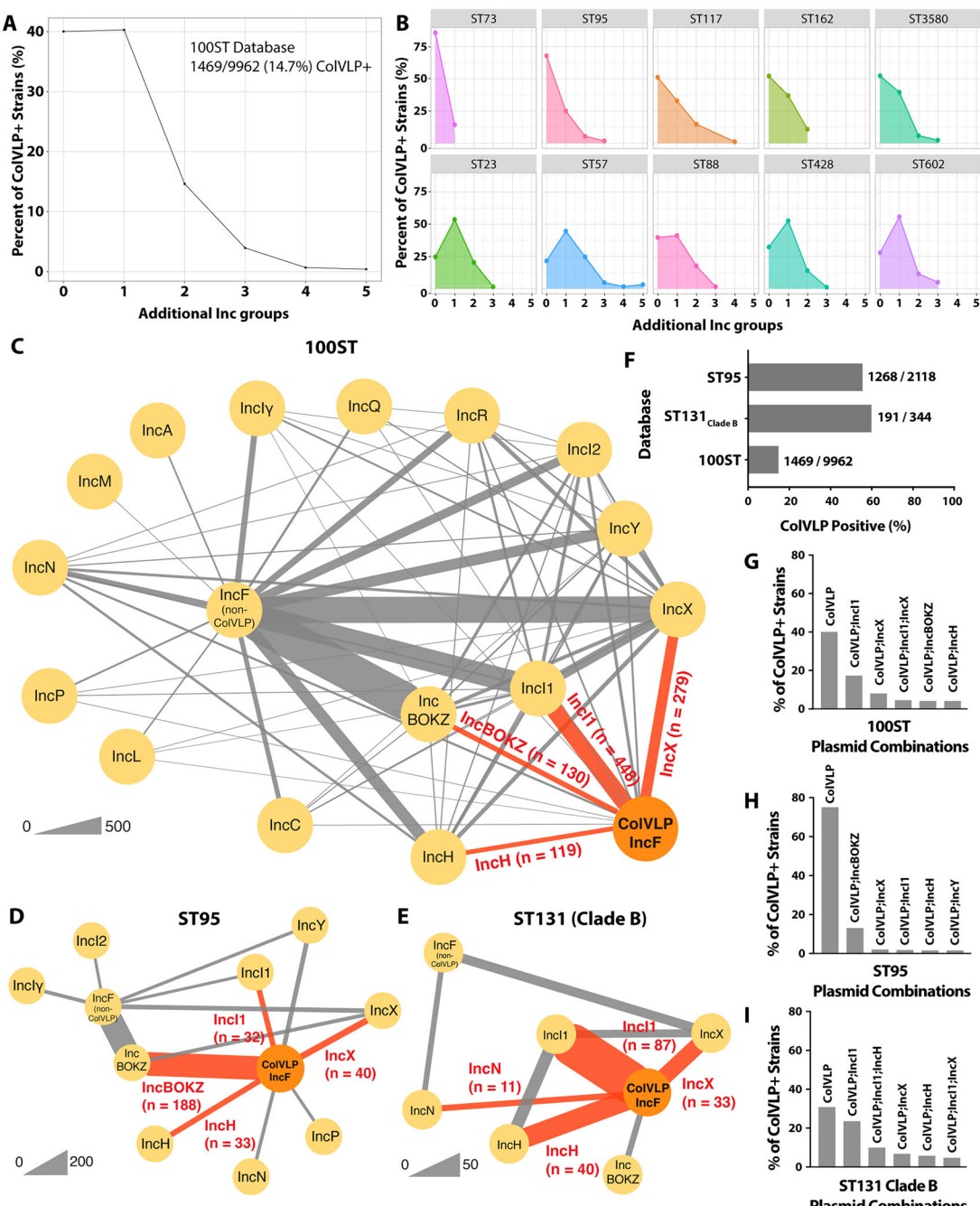

**Fig. 3 | ColV-like plasmids (ColVLPs) and co-resident plasmids in a 100 Sequence Type (ST) *E. coli* draft genome database.** Presence of additional replicons in ColVLP+ draft genomes of the **A** 100ST database and **B** top 10 ColVLP+ STs within the 100ST database. The 100ST database consists of 9969 draft *E. coli* genomes from the top 100 STs in Enterobase. Replicon co-carriage network in the **C** 100ST, **D** ST95 and **E** ST131 (Clade B) databases. The ST95 and ST131 (Clade B) databases consist of 2118 and 344 draft genomes, respectively. ColVLPs were plotted as a separate node from IncF plasmids (IncF non-ColVLP). Total plasmid co-carriage events ($n \geq 10$) were plotted using Cytoscape v3.9.1. Red lines denote the top four associations with ColVLPs. Line thickness between nodes is proportional to the number of co-carriage events. **F** Percent of ColVLP+ isolates in the 100ST, ST95 and ST131 Clade B databases. ColVLP+ isolates were tallied regardless of co-carriage events. Plasmid combinations of ColVLP+ isolates in the **G** 100ST, **H** ST95 and **I** ST131 (Clade B) databases. Plasmid combination frequency was normalised to the total number of ColVLP+ isolates in each database. The top six combinations of each database are shown. ColVLPs were identified using the criteria of Liu et al.[6]. Additional replicons were identified using ABRicate v0.8 against the PlasmidFinder[42] database using a 100% query length threshold. Figure acronyms: ST sequence type. Source data are provided as a Source data file in Supplementary Data 5,7 and 8.

protease (*ompT*) and the aerobactin auxiliary secretion factor (*shiF*)[4,50,51]. To investigate genetic differences between the chromosomal- and plasmid-encoded genes, we constructed phylogenetic trees of these virulence genes using *E. coli* chromosomal homologues identified from the NCBI RefSeq (1,377 chromosomes from completely sequenced strains) and their respective ColVLP-associated homologues from the

233 ColVLP database (Fig. 4A–F). For all virulence genes, most ColVLP-associated sequences clustered separately from their chromosomal homologues, indicating that the majority of ColVLPs carry a set of virulence loci distinct from that of their host chromosome.

We also noted rare instances where genes of ColVLP and chromosomal origin were interspersed in the phylogeny. To investigate

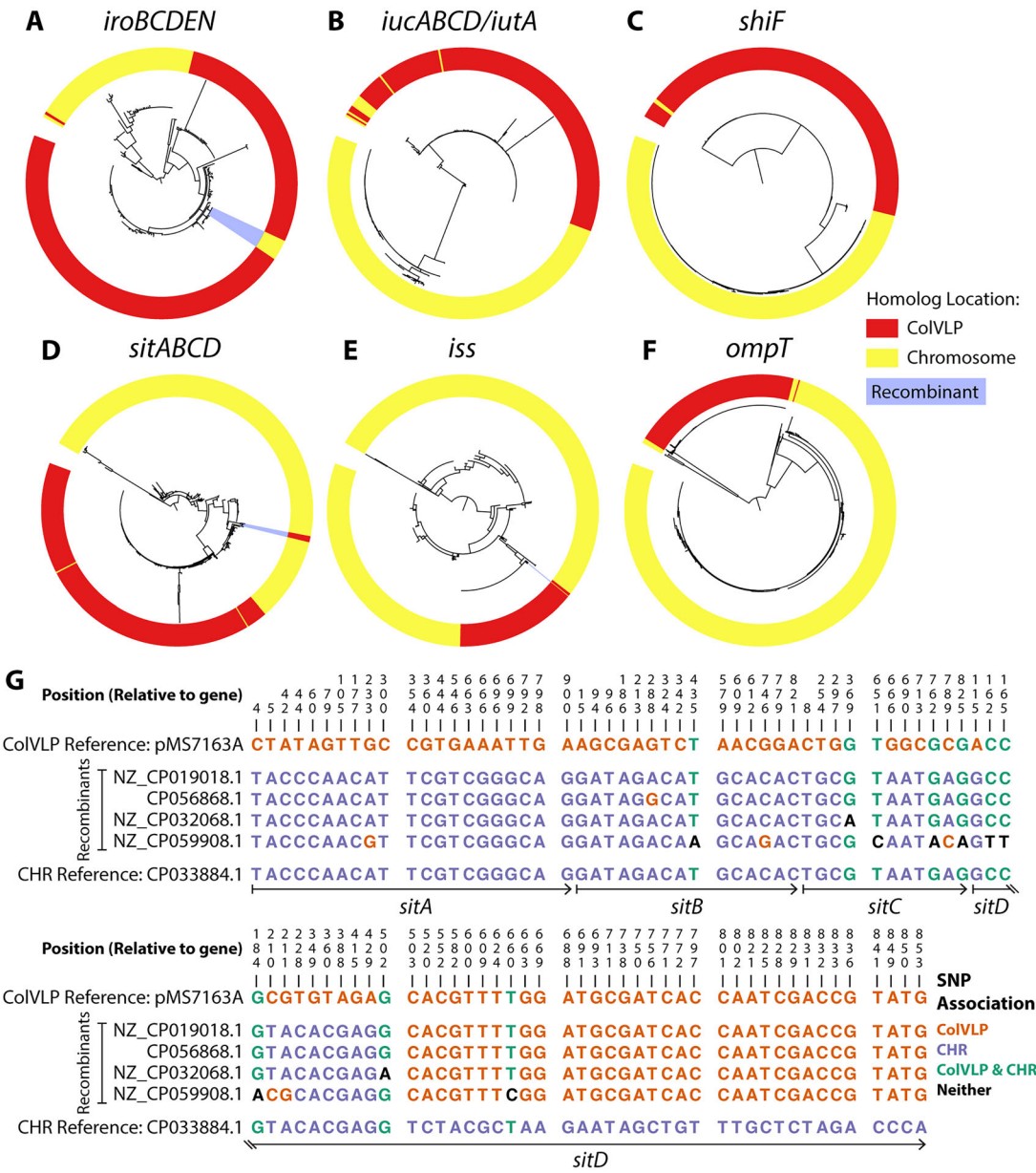

**Fig. 4 | ColV-like plasmid (ColVLP) virulence genes.** Phylogeny of the virulence genes **A** *iroBCDEN*, **B** *iucABCD/iutA*, **C** *shiF*, **D** *sitABCD*, **E** *iss* and **F** *ompT*. Nucleotide sequences were extracted from the NCBI RefSeq (1,377 *E. coli* chromosomes) and 233 ColV-like plasmid (ColVLP) collection, aligned using Clustal Omega, and visualised as a midpoint-rooted tree using FastTree and the interactive Tree of Life. Potential recombinant homologues are highlighted in blue. **G** Nucleotide identity (%) comparing the non-conserved sites of the recombinant *sitABCD* operons with these homologues for recombination, we manually checked the fol- representative chromosomal and ColVLP homologues. Alignments between the four potential recombinant *sitABCD* operons (from NZ_CP019018.1, CP056868.1, NZ_CP032068.1, NZ_CP059908.1) and the representative homologues (ColVLP - NZ_CP026854.1, Chromosome - CP033884.1) were performed using CLC Main Workbench. Figure acronyms: SNP single nucleotide polymorphism, CHR chromosome.

these homologues for recombination, we manually checked the following for possible recombination: (i) every ColVLP-associated sequence which had a long branch indicative of large sequence variation; and (ii) every sequence that did not cluster with its associated group.

For the virulence genes *iucABCD/iutA*, *shiF* and *ompT*, all outliers were confirmed to be the result of inactivating mutations (frameshift or internal stop) or suspected mis-assembly. For *iss*, only one outlier was found, where a chromosomal-associated copy was found to cluster with ColVLP-associated homologues—while possibly recombination, more sequencing data is needed to confirm this observation.

For *iroBCDEN* and *sitABCD*, the ColVLP-associated homologues with long branches were also confirmed to contain inactivating

mutations, and *sitABCD* additionally had two chromosomal homologues that clustered with ColVLP-associated copies due to suspected mis-assembly. Despite this, we were able to find convincing evidence of homologous recombination between ColVLPs and the chromosome. In the case of *iroBCDEN*, four fully intact operons that shared 99.97% nucleotide sequence similarity to a ColVLP-associated sequence were found on the *E. coli* chromosome (NZ_CP062985.1, NC_017632.1, NZ_CP012631.1, NZ_CP019777.1). More interestingly, with *sitABCD*, we found four ColVLP-associated sequences that formed their unique cluster within the chromosomal homologues. An alignment and comparison of non-conserved sites revealed that these four ColVLP-associated *sitABCD* sequences were hybrids between a ColVLP and chromosomal homologue, where the initial

section (*sitABC* and start of *sitD*) shared high conservation with the reference chromosomal sequence, and the latter section of *sitD* shared high conservation with the reference ColVLP-associated sequence (Fig. 4G).

Taken together, our results indicate that while most virulence loci carried on the ColVLP are distinct from their chromosomal counterparts, homologous recombination events can occur, resulting in switching of the entire operon (as evidenced by *iroBCDEN*) or even a partial section of a virulence gene (as evidenced by *sitD*) between ColVLPs and the *E. coli* chromosome.

## OmpT variants are functional but exhibit differences in substrate specificity

The phylogenetic distribution of the ColVLP-encoded and chromosomal-encoded virulence genes suggests conserved differences in their coding sequences. To explore if these differences altered function, we investigated OmpT, an outer membrane protease that cleaves antimicrobial peptides between two dibasic residues[52,53] and promotes resistance to human LL-37[54]. In *E. coli*, two main OmpT variants exist—the chromosomal-encoded OmpT and the ColVLP-encoded OmpTp (also referred to as ArlC)[19]. A query of 1377 complete *E. coli* genomes from NCBI RefSeq[26] revealed that ~65% of strains carry at least one *ompT*, with ~52.3% carrying a single copy on the chromosome and ~5% carrying at least one chromosomal and one plasmid copy (Fig. 5A). Notably, when the 100ST draft genomes was ranked for *ompT* carriage, >50% strains from the top five STs (ST428, ST117, ST23, ST88, ST141), and 39-50% of strains from the next five ranked STs (ST101, ST58, ST95, ST906, ST162), contained >1 *ompT* allele (Fig. 5B). These STs all belong to ExPEC/APEC pathotypes[6,7,9,43,45,55–59].

Using MS7163, an ST95 ExPEC reference strain that carries three *ompT* genes (two chromosomal-encoded alleles—*ompT1* and *ompT2*, and one ColVLP-encoded allele—*ompTp*)[32], we investigated the contribution of OmpTp to LL-37 resistance. The two chromosomal OmpT variants are highly similar (sharing 98.7% amino acid identity) and distinct from the plasmid variant (73.2% amino acid identity to either chromosomal copy) (Supplementary Fig. 13). A comparison of the active, catalytic and LPS-binding sites between OmpT1/OmpT2 and OmpTp revealed that all key residues were conserved, except for the two LPS-binding sites at residues 175 (R175H) and 226 (K226N) (Fig. 5C and Supplementary Fig. 13). Superimposition of the predicted OmpT1/2/p structures generated in ColabFold revealed almost identical structures, comprising a beta-barrel that extends from the outer membrane to the extracellular space (Fig. 5C–E).

To test for the activity of each OmpT variant, each individual allele was PCR amplified and cloned into the low-copy-number pSU2718-Km expression vector. This generated a series of constructs containing each individual *ompT* allele placed under the control of an identical promoter and ribosomal binding site. These resultant constructs were then used to transform an *E. coli* K-12 MG1655 mutant deleted for *ompT* (i.e. MG1655 *ompT*::Cm), with two assays subsequently performed to test the function of each *ompT* allelic variant. First, we used fluorescence resonance energy transfer (FRET) to measure the cleavage of a fluorescently labelled synthetic peptide derived from the sequence of complement C2, which revealed the greatest activity for OmpTp followed by OmpT1, but no detectable activity for OmpT2 (Fig. 5F). Second, we tested the ability of each OmpT variant to cleave LL-37 by utilising LC-MS to track a predicted LL-37 cleavage product. All OmpT variants were able to cleave LL-37 (Fig. 5G). Taken together, this data reveals that the ExPEC strain MS7163 encodes for three functional OmpT variants that exhibit variable substrate specificity, with OmpTp displaying activity against two different peptides.

## The ColVLP OmpTp variant contributes to MS7163 LL-37 resistance

To investigate the contribution of the ColVLP-encoded OmpTp variant to LL-37 resistance, MS7163 *ompT* mutants expressing one, two or no active OmpT variants were generated using λ-Red-mediated homologous recombination. The survival of each mutant was examined after 2 h incubation with LL-37 in PBS. In these experiments, survival of the MS7163 triple mutant was significantly attenuated compared to the WT parent strain ($p < 0.0001$; Fig. 5H), demonstrating that the combined absence of all three OmpT variants enhances susceptibility to LL-37. The expression of any single OmpT variant did not restore LL-37 resistance to WT levels (OmpTp [Fig. 5H]; OmpT1 or OmpT2 [Supplementary Fig. 14A]), even when expressed from a plasmid expression vector (Supplementary Fig. 14B), revealing the requirement for a cooperative mode of resistance involving multiple OmpT variants. Despite this, the expression of OmpT1 and OmpTp consistently increased LL-37 resistance without always reaching statistical significance (Supplementary Fig. 14A; Fig. 5H). Notably, the co-expression of OmpTp+OmpT1 restored LL-37 resistance to a level comparable to WT (Fig. 5H), but this was not observed for the co-expression of OmpTp+OmpT2 nor OmpT1+OmpT2 (Supplementary Fig. 14A). Taken together, the data demonstrate a role for the ColVLP-encoded OmpTp variant in resistance to LL-37.

## Discussion

Here we report on the convergence of virulence and antibiotic resistance in plasmids associated with ExPEC strains that cause human and animal infections. By generating a detailed ColVLP cladogram using an ORF-based binarised structure network analysis tool[30], we identified clusters of closely related virulence plasmids and co-integrate plasmids that harbour high AMR gene burden. Furthermore, we show that the ColVLP-encoded OmpTp protease contributes to ExPEC resistance to LL-37, providing experimental evidence to link its function to enhanced virulence.

To date, most studies tracking ColVLP evolution have done so by tracing the acquisition and subsequent modification of mobile genetic elements carrying AMR determinants in a group of closely related FII-2/FIB-1-containing ColVLPs[22,37]. These molecular signatures include the presence of a class 1 integron truncated by IS26[37], the acquisition of a Tn1721/Tn21 hybrid transposon, the acquisition and in situ modification of Tn1721, as well as the presence of Colicin Ia[22]. While these approaches allow for high-resolution understanding of plasmid evolution, they focus on sub-lineages of FII-2/FIB-1 ColVLPs (part of cluster 6; Supplementary Fig. 15). Here, by utilising an ORF-based approach independent of genetic markers and a conserved backbone, we broaden these existing analyses by providing a comprehensive, evolution-independent overview of currently sequenced ColVLPs (clustered by gene content). Our analysis identified diverse clusters of ColVLPs, providing a framework to characterise their contributions to ExPEC pathogenesis. We do note, however, that there are several limitations to our binary clustering methodology[30]. The methodology is annotation-dependent; independently acquired antibiotic resistance genes may influence clustering, and non-related plasmids may be clustered closer together than expected due to many shared 0's between pairwise alignments that indicate no homology. Given these limitations, we suggest this method is not optimal for inferring plasmid evolution, but instead is better suited to clustering annotated and related plasmids, such as ColVLPs.

Our study revealed that ColVLPs are largely restricted to *E. coli* and are only rarely found in *Klebsiella pneumoniae* or *Salmonella enterica*. Given the conjugative nature of some ColVLPs, their restriction to *E. coli* suggests that ColVLPs have adapted to their *E. coli* hosts and may not be stably maintained when acquired by other Enterobacteriaceae. Despite this, there is a remarkable similarity between

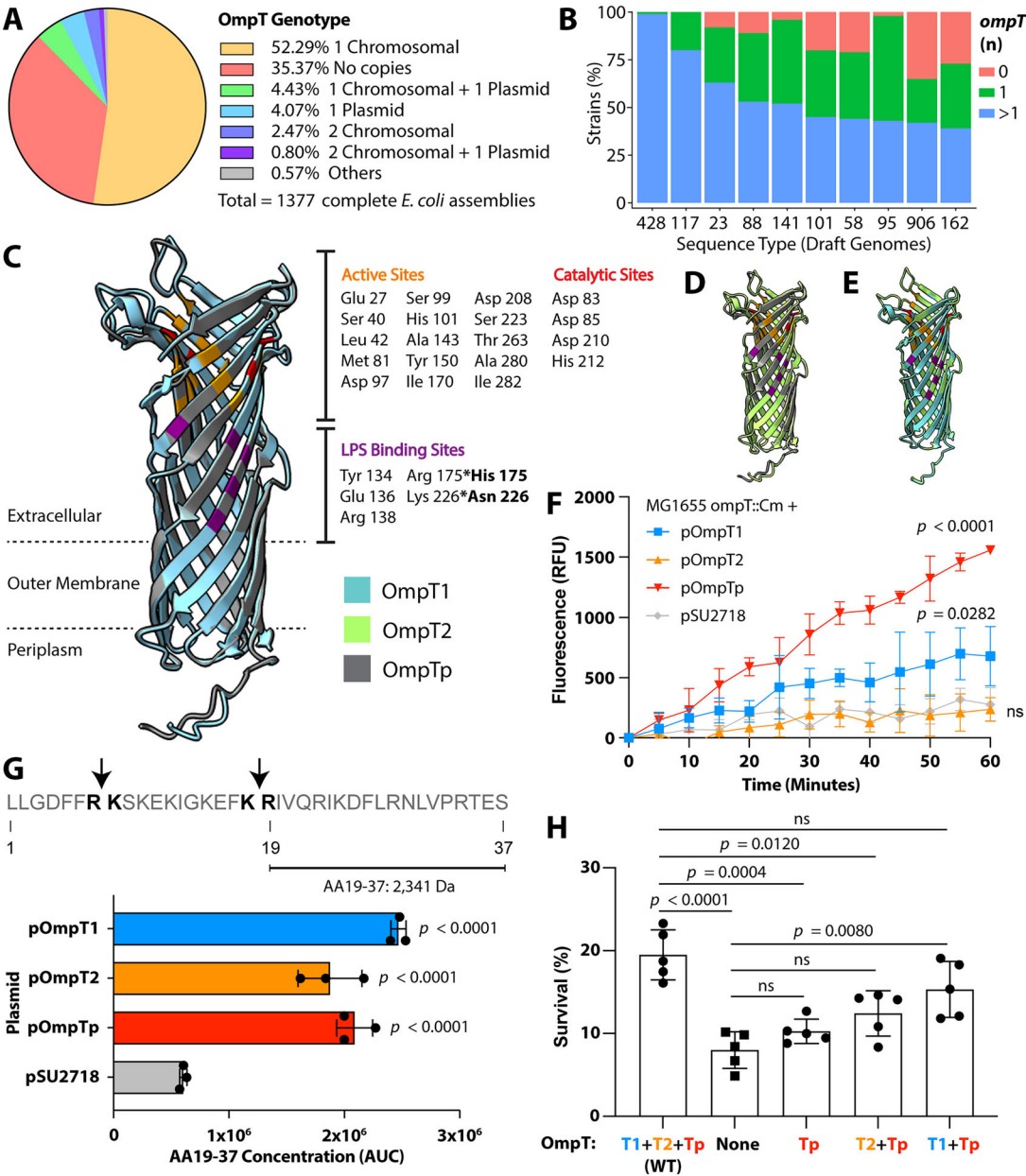

**Fig. 5 | Extra-intestinal pathogenic *E. coli* OmpT variants. A** Predicted *ompT* genotype of 1377 *E. coli* assemblies. Prevalence of *ompT* in chromosomal and plasmid contigs was determined using the MS7163 *ompT1* nucleotide sequence as a BLASTn query against 1377 complete *E. coli* genomes from NCBI RefSeq at a 65% query length and alignment threshold. **B** Top 10 STs carrying >1 *ompT* copy. Prevalence of *ompT* in the 100ST *E. coli* draft genomes database was determined using the MS7163 *ompT1* nucleotide sequence as a BLASTn query. Superimposition of predicted **C** OmpT1 and OmpTp, **D** OmpT2 and OmpTp, **E** OmpT1 and OmpT2 protein structures from MS7163. Protein structures were predicted using ColabFold and annotated using ChimeraX. **F** FRET complement C2-derived peptide cleavage over time by MG1655 *ompT*::Cm expressing OmpT1, OmpT2, or OmpTp. All readouts were normalised against time 0 fluorescence. Data is presented as Mean ± SD of three biological replicates. One-way ANOVA with Dunnett's multiple comparisons against the MG1655 *ompT*::Cm + pSU2718 control were performed on endpoint values. **G** Liquid chromatography–mass spectrometry (LC-MS) analyses of OmpT-mediated LL-37 cleavage. The LL-37 sequence is shown with predicted OmpT cleavage sites as a bold arrow. The bar graph shows the concentration of LL-37$_{19-37}$ after incubation with MG1655 *ompT*::Cm + pOmpT1/2/p strains, quantified using LC-MS. Data are presented as Mean ± SD of three biological replicates. Significance was determined using a One-way ANOVA with Dunnett's multiple comparisons against the pSU2718 vector-only control. **H** Survival of MS7163 derivatives against 0.05 mg/mL LL-37. Data are represented as Mean ± SD of five biological replicates. Significance was determined using a One-way ANOVA with Dunnett's multiple comparisons against the WT strain. Survival of strains after 2 h of incubation with LL-37 is expressed as a percentage of the CFU count in PBS control at time 0. Figure acronyms: LPS lipopolysaccharide, WT wildtype, AA amino acid, AUC area under the curve. Source data are provided as a Source data file in Supplementary Data 5 and 6.

ColVLPs of ExPEC and virulence plasmids of hypervirulent *Klebsiella pneumoniae* (hvKp)[60]. Both plasmid types carry virulence genes encoding the salmochelin and aerobactin siderophore systems[61] that contribute to the pathogenicity of their hosts[17,62]. In contrast, some virulence genes remain unique to each plasmid, such as *rmpA* (located on the hvKp virulence plasmid), whose expression enhances capsule production and results in a hypermucoid phenotype, and *ompT* (located on ColVLPs). In line with our findings, several completely sequenced hvKp virulence plasmids contain AMR genes likely acquired through recombination (mediated by IS elements) or co-integration[63–65]. It

remains to be seen if a similar convergence of AMR and virulence determinants exists in hvKp virulence plasmids. Such convergence is alarming not only because of the increased impact these plasmids can have on both pathogenicity and resistance, but also because antibiotic use could select for hypervirulent strains harbouring these plasmids.

The identification of ColVLP co-integrates reveals a mechanism to explain how ColVLPs can rapidly expand their AMR repertoire and host range, a phenomenon observed in other studies[66–68]. While some studies demonstrate co-integration as a transient event between co-resident plasmids[69], stable clinically-relevant plasmid co-integrates have been described, such as the enterotoxigenic *E. coli* virulence plasmid pCoo—a co-integrate between an IncF and an IncI1 plasmid[70]. The independent identification, sequencing and assembly of closely related *mcr-1*-positive co-integrates (Fig. 2) strongly suggests stability or repeated co-integration rather than transience, although this remains to be experimentally validated. Regardless, the identification of these *mcr-1*-positive ColVLP co-integrates from human ExPEC strains warrants deeper investigation through the implementation of unbiased (long-read) sequencing to capture the full extent of their distribution and transmission. Indeed, the problem of incomplete plasmid assemblies is a limitation of our plasmid replicon co-carriage analyses. Although our data show that ColVLP+ strains can carry additional replicons (IncI1, IncX, IncB/O/K/Z and IncH as the most common), we cannot differentiate between independent co-occurrence or co-occurrence on the same plasmid backbone without closed assemblies.

An intriguing aspect of our analyses was the observation that some ColVLP-encoded virulence loci have recombined with their host chromosome, resulting in full operon switching (*iroBCDEN*; Fig. 4A) to hybrid sequences (*sitABCD*; Fig. 4G). Here, our database consisted of completely sequenced *E. coli* genomes irrespective of ColVLP carriage and is likely predominantly from strains that do not normally carry ColVLPs. As recombination between the ColVLP and the chromosome can only occur when a strain carries a ColVLP, we suspect these recombination events occur at a rate higher than what our results suggest. To fully elucidate the extent of recombination, a large collection of completely sequenced genomes belonging to ColVLP+ STs is required, but is currently lacking.

Prior studies have suggested that the presence of multiple OmpT variants in the same strain confers a fitness advantage by expanding substrate range and protease activity against antimicrobial peptides, including LL-37 and RNase 7, that contribute to innate defence in the urinary tract[19,71]. Our results support the existence of distinct substrate preferences for each OmpT variant, noting also that each variant could effectively cleave LL-37 following plasmid-mediated expression in the K-12 host strain MG1655 (Fig. 5G). An important advance in our study was the demonstration that the expression of OmpTp and OmpT1 at wild-type levels enhanced protection of the ExPEC strain MS7163 against LL-37 (Fig. 5H), providing direct evidence that the ColVLP-encoded OmpTp plays a role in virulence. An underexplored aspect of OmpT-mediated LL-37 cleavage is the putative requirement of LPS for OmpT activity[72]. Although a recent study demonstrated that mutation of the predicted LPS-binding sites did not abolish OmpT activity[73], this does not rule out the possibility for LPS non-specific binding or additional LPS-binding sites. While we identified two residues involved in OmpT-LPS binding that differed between OmpT1/OmpT2 and OmpTp (R175H and K226N), the impact of these amino acid changes on substrate specificity remains to be determined. Overall, the findings from our study, together with a previous report that showed OmpTp can cleave the antimicrobial peptide RNase 7[19], demonstrate that the ColVLP-encoded OmpTp protease functions in tandem with its OmpT chromosomal counterparts to enhance ExPEC survival in the presence of soluble innate immune effectors.

In summary, we have detailed the genetic characteristics of ColVLPs. These plasmids function as a horizontally disseminated vehicle to enhance the virulence armoury of ExPEC through the production of factors that lead to increased iron acquisition via siderophores, metal ion transport, serum resistance and the cleavage of host antimicrobial peptides. By demonstrating the carriage of multiple AMR genes on these plasmids, including co-integrate ColVLPs that harbour multiple replicons, our findings expand the role of ColVLPs in the plasmid-driven convergence of virulence and resistance in ExPEC.

## Methods

### Bacterial strains and growth conditions
All strains were routinely cultured at 37 °C in either lysogeny broth (LB) or N-minimal medium (50 mM Bis-Tris, 5 mM KCl, 7.5 mM $(NH_4)_2SO_4$, 0.5 mM $K_2SO_4$, 0.5 mM $KH_2PO_4$, 0.1% casamino acids) supplemented with 0.2% glucose and 1 mM $MgCl_2$ (NMM) under shaking (250 rpm) or static conditions. Antibiotics were supplemented at the following concentrations as appropriate: chloramphenicol (30 µg/mL); kanamycin (100 µg/mL), gentamycin (20 µg/mL). All strains were stocked at −80 °C in 15% glycerol. A full list of strains and plasmids used is available in Supplementary Data 2. MS7163 is a reference ST95 ColVLP+ ExPEC strain which was isolated from a patient with acute pyelonephritis and belongs to the virulent O45:K1:H7 clone that causes one-third of all neonatal meningitis cases in France[32].

### PCR, transformations and mutant generation
Colony PCRs were performed using OneTaq DNA polymerase (New England Biolabs). Sanger sequencing reactions were prepared using BigDye Terminator Mix v3.1 and sequenced by the Genetic Research Services, UQ. Electrocompetent cells were prepared, and transformations were performed as previously described[74]. All mutants were constructed using λ-Red-mediated homologous recombination as previously described[74]. All constructs were confirmed by Sanger sequencing. DNA fragments for cloning and mutations were amplified using KAPA HiFi polymerase (Roche). All primers were synthesised by IDT (Singapore), and a full list of primers used is available in Supplementary Data 3.

### Construction of 233 ColVLP database
ColVLP-associated loci from the ColV plasmid pMS7163A[32] were used as a BLASTn (v2.14.0+) query against the NCBI RefSeq[26] (21,034 chromosomes; 19,736 plasmids; 02/01/2021) and PLSDB[25] (34,513 plasmids; 23/06/2021) databases to identify a non-redundant collection of 233 ColVLPs as described in Liu et al.[6]. The NCBI RefSeq contained the complete genomes of 820 *Klebsiella spp.*, 930 *Salmonella spp.*, and 1441 *Escherichia spp.* isolates. Incomplete plasmids with ambiguous nucleotides were removed from further analyses. ColVLPs were assigned to pre-defined PTUs using COPLA[75], using both linear and circular topologies with default options.

### Cladogram and network construction
All unique ORFs from the constituent plasmid sequences were concatenated into a single hypothetical plasmid. The hypothetical plasmid was subsequently compared against each plasmid in a sequence similarity search, generating a binary ORF presence/absence sequence[30]. All binary sequences were subject to hierarchical clustering based on Manhattan distance using the stats package in the R (v4.3.2) environment[76], and visualised as a midpoint-rooted cladogram using the interactive Tree of Life[77]. Cladogram clusters (>1) were determined using the total sum or square method. Split decomposition networks were generated using the ORF-based binarised structure network analysis of plasmids tool[30], and visualised using SplitsTree (v6.3.10)[78]. Network clusters were determined with a Dice index threshold of >0.71.

## General bioinformatic analyses

Pairwise alignments were performed using BLASTn[79] and visualised with EasyFig (v2.2.2)[80] with a minimum coverage length of 500 bp. Presence/absence of ColVLP-associated genes was determined using pMS7163A sequences[32] as a BLASTn query at a 95% alignment length and 90% percent identity threshold. Operons were only considered present if all constituent genes were identified. Conjugative ability was predicted based on the presence of all genes required for F plasmid conjugation[81] using a BLASTn search at a 90% alignment length and percent identity threshold. F plasmid replicons were determined using a query against the IncF pMLST scheme[28,82] with an exact match threshold. Resistance genes and non-F plasmid replicons were determined using ABRIcate (v0.8.10) (https://github.com/tseemann/abricate) against the ARG-ANNOT[83] and PlasmidFinder[41] databases, respectively, at a 100% query length threshold. Host *E. coli* STs were determined using the MLST (v2.23.0) tool (https://github.com/tseemann/mlst) and the PubMLST database[82]. All features are available in Supplementary Data 4.

## Identification of ColVLP co-integrates

ColVLP-associated loci from the ColV plasmid pMS7163A[32] were used as a BLASTn query against NCBI (query performed 4th April 2024) as described in Liu et al.[6] to identify ColVLPs. Incomplete plasmids with ambiguous nucleotides and partial sequences were removed from further analyses. Plasmid replicon queries from PlasmidFinder[41] were used as a BLASTn query against the ColVLPs at a 90% identity and 100% alignment length threshold to identify non-F replicons.

## Plasmid carriage inference and plasmid co-carriage network generation

ColVLP and plasmid carriage were inferred using the criteria of Liu et al.[6] and PlasmidFinder[41], respectively, for the following draft genomes databases: (i) 100ST database consisting of ~100 random strains from the top 100 STs in Enterobase (downloaded on 18th December 2020)[49,84]; (ii) ST95 database consisting of 2118 strains from Enterobase (downloaded on 24th May 2021)[84]; (iii) ST131 database consisting of 3857 strains from Enterobase (downloaded 17th July 2018)[84], which was additionally separated into their clades as described previously[85]. All predicted co-carriage events ($n > 10$) were quantified and plotted as a network using Cytoscape (v3.9.1)[86]. ColVLPs and IncF plasmids were plotted as separate nodes (i.e., ColVLP positive draft genomes were considered as F plasmid negative).

## Virulence gene phylogenetic analyses

Virulence genes were identified from the NCBI RefSeq[26] *E. coli* chromosomal database (1,377 *E. coli* chromosomes; 02/01/2021) and the 233 ColVLP databases using the loci from pMS7163A[32] as a BLASTn query at a 60% alignment length threshold. Nucleotide sequences were aligned using Clustal Omega (v1.2.3)[87], built into a phylogenetic tree using FastTree (v2.1)[88], and visualised as a midpoint-rooted phylogenetic tree using the interactive Tree of Life[77].

## OmpT homologue comparison

OmpT prevalence was determined using MS7163 *ompT1* (MS7163_RS03130)[32] as a query against the NCBI RefSeq database (1377 complete assemblies; 02/01/2021) at a 65% query length and alignment threshold. Predicted protein structures were generated using ColabFold (v1.5.2)[89] and annotated using ChimeraX (v1.6.1)[90]. Amino acid sequence alignments were performed and visualised using CLC Main Workbench (v23.0.4) (QIAGEN).

## FRET assay

The synthetic FRET peptide [2Abz-SLGRKIQI-K(Dnp)-NH₂] with ortho-aminobenzoic acid (Abz) as the fluorescent and group 2,4 nitrophenyl (Dnp) as the quencher was purchased from Mimotopes at a purity of >95% and resuspended in sterile Milli-Q water. The FRET protocol was adapted from Thomassin et al.[91]. Briefly, bacterial cells were grown to mid-log phase in NMM and standardised to an $OD_{600}$ of 0.5 in PBS. Bacterial cells were mixed with the FRET peptide at final concentrations of $3 \times 10^8$ CFU/mL and 3 μM (in PBS), respectively, and incubated at 23 °C for 60 min, with fluorescence observation (excitation wavelength 325 nm, emission wavelength 430 nm) every 5 min. Data were normalised to time 0 readouts.

## LL-37 Proteolytic cleavage and LC/MS quantification

LL-37 peptides (LLGDFFRKSKEKIGKEFKRIVQRIKDFLRNLVPRTES) were purchased from Mimotopes and resuspended in sterile Milli-Q water. Bacterial cells were grown to mid-log phase in NMM and standardised to $OD_{600}$ 1.8. Bacterial cells were mixed with LL-37 in PBS, at final concentrations of $7 \times 10^7$ CFU/mL and 0.1 mg/mL, respectively, and incubated at 37 °C for 2 h. After incubation, trifluoroacetic acid was added to a final concentration of 1%. Bacterial cells were pelleted through centrifugation, and the supernatant was filtered through a 0.22 μm PVDF filter (Merck; SLGV033RS). Samples were stored at −80 °C prior to being analysed by LC-MS on a Shimadzu Nexera, LC30, uHPLC (Japan) coupled to a Triple Tof 5600 mass spectrometer (ABSCIEX, Canada) equipped with a duo electrospray ion source. Samples (20 μL) were injected onto a 2 × 150 mm Phenomenex, Jupiter, 300A, 3.5 μm C18 column (Phenomenex, Australia) at 200 μl/min. Peptides were eluted using linear gradients of 2–60% solvent B over 60 min, followed by 60–80% solvent B over 2 min, and finally 80-98% solvent B in 0.5 min. Solvent A consisted of 0.1% formic acid (aq) and solvent B contained acetonitrile/ 0.1% formic acid (aq). The ionspray voltage was set to 5500 V, declustering potential (DP) 100 V, curtain gas flow 25, nebuliser gas 1 (GS1) 50, GS2 to 60, interface heater at 150 °C and the turbo heater to 500 °C. A 400 ms full scan TOF-MS spectrum over the mass range 500–2000 $m/z$ was acquired and processed using Analyst TF 1.6 software (ABSCIEX, Canada). Quantitation of products corresponding to LL-37$_{AA19-37}$ was performed using MultiQuant 3.0.3 (ABSCIEX, Canada). The mass spectrometry proteomics data have been deposited to the ProteomeXchange Consortium via the PRIDE[92] partner repository with the dataset identifier PXD060817.

## LL-37 time-kill assay

Bacterial cells were grown in N-minimal medium to mid-log and standardised to an $OD_{600}$ of 1.8 in PBS. Bacterial cells were mixed with LL-37 (in PBS) at final concentrations of ~$4 \times 10^8$ CFU/mL cells and 0.05 mg/mL, respectively, and incubated at 37 °C for 2 h. After incubation, bacterial cells were serially diluted, spotted on LB agar, and enumerated the next day. Data is represented as a percentage of the CFU count in the PBS control at time 0.

## Reporting summary

Further information on research design is available in the Nature Portfolio Reporting Summary linked to this article.

## Data availability

The LC-MS data generated in this study have been deposited in the ProteomeXchange Consortium via the PRIDE[92] partner repository under the accession code PXD060817. The metadata for the 233 ColVLP, 100ST, 1377 NCBI RefSeq complete genomes, ST95 draft genomes, and ST131 Clade B draft genomes databases generated in this study are provided in the Source data. Source data are located in the Supplementary Data files. Sequence data used to verify plasmid constructs is available from the authors upon request. All other data generated in this work are presented in the manuscript.

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

## Acknowledgements

The authors thank the University of Queensland Institute for Molecular Bioscience colleagues Nicholas Yuen for providing technical assistance and Alun Jones for advice and assistance with LC/MS quantification of the LL-37 cleavage products. This research was supported by use of the Nectar Research Cloud and by QCIF. The Nectar Research Cloud is a collaborative Australian research platform supported by the NCRIS-funded Australian Research Data Commons (ARDC). The work was supported by grants from the Australian National Health and Medical Research Council (APP2001431 to M.A.S., M.-D.P. and N.T.K.N.; APP2037698 to M.A.S., M.-D.P.), the Australian Medical Research Future Fund (MRFGH000028 to M.A.S., M.-D.P.) and the Australian Research Council (DP230101930 to M.A.S.).

## Author contributions

Conceptualisation: Z.J.L., N.T.K.N., M.-D.P., M.A.S. Investigation: Z.J.L., N.T.K.N., C.R., C.C., I.M.-R. Formal analysis: Z.J.L., N.T.K.N., M.-D.P., M.A.S. Supervision: M.-D.P., M.A.S. Writing—original draft: Z.J.L., M.-D.P., M.A.S. Writing—review and editing: all authors. All authors read and approved the final manuscript.

## Competing interests

The authors declare no competing interests.
