## [Peer Review file · Nature Communications]

Convergence of plasmid-driven virulence and antibiotic resistance in *Escherichia coli*

Corresponding Author: Professor Mark Schembri

Version 0:

Reviewer comments:

Reviewer #1

(Remarks to the Author)

The manuscript, "Convergence of plasmid-driven virulence and antibiotic resistance in *Escherichia coli*", by Lian et. al reports interesting findings on the composition of ColVLP plasmids, their diversity/similarity, co-occurrence with additional plasmids within top 100ST *E. coli* hosts, and the incidence of cointegration of ColVLP with some of these co-resident plasmids. The significance of the cointegration of ColVLP plasmids with additional plasmids leads to the potential for convergence of antibiotic resistance and virulence factors within the same plasmid and the study culminates in the evaluation of the virulence associated antimicrobial peptide activity/specificity of the unique plasmid encode protease variant OmpTp as compared to the chromosomally encoded variants OmpT1 and OmpT2 found in the ST95 representative strain MS7163. This is evaluated by assessing the in vitro proteolysis of potential substrates, which could be encountered in a human host, using a heterologous *E. coli* host and survival of MS7163, and various OmpT mutant combinations, to LL-37 (human cathelicidin). This study is well written and nicely outlines its findings with informative and well-organized figures, and I wish to let the authors know this was a pleasure to read. As a testament to this observation, I have very few critiques to provide and suggest them as potential points for improving clarity.

Line 127 (Fig 1 legend) Specify "(imm)" abbreviation, presumably it's immunity

Line 170, is the third Cluster 1 plasmid, that does not encode *mcr-1*, so substantially different from the other hybrid plasmids that it could not be included in the alignment with the pMS7163A and pSTM6-275 references? Its conspicuous absence has puzzled me, and a line of justification for its exclusion would be helpful.

Lines 206 – 210, These three sentences are confusing as the IncH12+ isolates are not all of human origin. As the information in the first sentence appears redundant and inaccurate, consider consolidating the three sentences on these lines, discussing IncH12+ & IncN+, into two sentences.

Line 228, referencing Fig 3A (rather than 3F) would be sufficient, as referring to Fig 3F this early distracts from the initial analysis being teed up for Fig 3A as it contains the 1469/9962 population data and 14.7%. Additionally, jumping to Fig3F here is tricky because 2/3 of the data in Fig3F is not explained until Line 263, so this leads to additional hunting and distracts from the very linear and organized story of the paper thus far.

Fig 3B., it was difficult to find the ST types referenced in lines 232 – 234 as the organization of the plasmid occupancy % graphs for each ST type appears random (likely based on Top1-10 ST types). It would be easier to interpret and appreciate the dichotomous plasmid carriage of the ST types (ColVLPs alone vs. +additional plasmids) if they were organized with the upper row of graphs being of one type in numerical ST order and the lower row being the other type in numerical ST order. As it stands, it's hard to appreciate at a quick glance how the two populations are indeed unique from one another, and I found myself having to hunt for each ST type referenced (lines 232-234) among the 10 plots of Fig 3B.

Fig 5H., Complementation of the *ompT*-null strain MS23578 (MS7163 Δ ompT2 Δ ompTp *ompT1*::Cm) with each of the individual *ompT* containing plasmids used for the proteolysis assays in MG1655 would be helpful in evaluating potential polar effects resulting from the introduction of these mutations. While the dose of the pOmpT encoded protease would likely exceed native levels, presumably the levels of each *ompT* variant on the pOmpT plasmid would be similar in this *ompT*-null strain and its inclusion would allow for survival-based contributions in their native host and pair nicely with the non-native

MG1655 data presented in Fig 5G.

Reviewer #2

(Remarks to the Author)

Review for Convergence of plasmid-driven virulence and antibiotic resistance in *Escherichia coli*

I enjoyed reading this one, the authors present on the genetic variation and possible gene exchange of virulence factors between ColV (and similar) plasmids.

Comments

* introduction is well described.

* The authors use a method of clustering plasmid sequences according to presence/absence of their expected gene list. This should work for the purposes used here, but I am curious why existing methods for plasmid taxonomy e.g. COPLA were not used. Admittedly, pMLST was used here. I usually see plasmid sequences compared with ANI or similar using the proportion of aligned sequence rather than the presence/absence approach used here. It could be that there is no real sequence similarity to make an alignment? Please clarify/justify this approach. See review of approaches in <https://www.sciencedirect.com/science/article/pii/S0147619X2300015X> or similar existing work.

* as a corollary to that, i am curious which genes are informing those clusters as many of the virulence genes (fig 1 - blue) seem at odds with it. On the other hand, these plasmids look like mosaic structures with genes having different histories of evolution and exchange, and clustering on the presence/absence pattern on the whole might not mean very much.

* Figure 1, perhaps a different colour coding for the different categories. Difficult to distinguish which column is host vs replicon vs conj. act. as they all use Red.

* Co integration result is interesting. Looks good.

* Checking against other genomes of STs of interest will be easier in future with datasets such as "all the bacteria" <https://www.biorxiv.org/content/10.1101/2024.03.08.584059v3>

* Discussion : "Our study revealed that ColVLPs are largely restricted to *E. coli*, and only rarely found in *Klebsiella pneumoniae* or *Salmonella enterica*" I did not see such a result in Results. Perhaps I missed it.

* Methods (Bioinformatics - the part I know) look fine. Meets my standards.

* You might get similar results with faster run times using USEARCH to replace BLAST and MAFFT to replace ClustalOmega.

Reviewer #3

(Remarks to the Author)

This manuscript describes the characterisation of a group of *E. coli* plasmids that carry both virulence and antibiotic resistance determinants. While the group of plasmids has been associated with virulence in *E. coli* that cause human and animal infections since the 1960s, examples reported over the last 20 years have acquired antibiotic resistance genes, representing a concerning convergence of clinically relevant phenotypes. The first half of this manuscript outlines an investigation of 233 complete plasmid sequences, and the second half describes the detailed characterisation of a plasmid-encoded outer membrane protease.

The characterisation of the outer membrane protease, OmpTp, is novel work that clearly defines its activity and has relevance to wider understanding of OmpT substrate specificity. The findings presented here suggest that OmpTp might contribute to *E. coli* virulence by conferring resistance to the human antimicrobial peptide cathelicidin (LL-37), which plays a protective antibacterial role in the urinary tract. This adds to the repertoire of well-characterised virulence determinants associated with this plasmid group, further strengthening their association with *E. coli* pathogenicity.

Sequence analysis includes: 1) clustering plasmids using an ORF-based approach and comparing cluster gene content, 2) screening plasmids for non-F-type replicons to explore cointegrate formation, 3) screening host genomes to describe co-resident plasmid types, and 4) comparing plasmid-associated virulence genes and their chromosomal homologues to assess the extent to which they recombine. This analysis has led to some interesting observations, particularly around the formation of cointegrates by members of this group. However, there are several issues with the analyses and conclusions

drawn from them.

One issue significantly detracting from the analysis is the disconnect between it and the existing literature on this plasmid group. With the literature considered, it appears that some methods employed here are not suitable for obtaining insights that advance the understanding of these plasmids. For example, previous work has characterised the molecular events associated with the acquisition of AMR determinants, describing differences that delineate sub-lineages with much greater resolution than ORF-based clustering can provide.

While the OmpTp work will be a valuable addition to the literature, the sequence-based analyses require reconsideration. I have provided more detailed comments below.

Comments on sequence analysis:

1) Plasmid clustering

The ORF-based clustering approach does not seem to be a stringent method for assessing plasmid relationships, particularly when complete plasmid sequences are available, as its simple presence/absence evaluation is blind to evolutionary events that shape and can be used to contextualise differences between plasmids. I have questions about the biological and evolutionary relevance of the clusters, and their overall pertinence given existing knowledge of ColV plasmid sub-lineages.

1a) Could a plasmid in cluster 1 or 2 be a deletion derivative of one in clusters 3-6? If so, a single molecular event might be having an overly significant impact on the shape of this cladogram, and clusters 1/2 might be artifacts.

1b) Lines 145-153: It appears that antibiotic resistance genes would have been included in the ORFs that informed this cladogram. If that is the case, are plasmids with large numbers of ARGs clustered together because they carry these ARGs? Do these clusters reflect evolutionary relationships, or have plasmids with convergent evolutionary trajectories (acquired the same genes in different events) been conflated?

1c) Collapsing resistance gene data to phenotypic summaries is non-ideal when considering the evolution of these plasmids, as it masks data that is crucial when considering the convergence of virulence and resistance. For example, sulphonamide resistance can be conferred by *sul1*, *sul2*, or *sul3*, each of which has been mobilised independently and has a distinct evolutionary history. The rarer *sul3* has been reported in a ColV plasmid, with the mechanism for its acquisition described (see PMID 26855083). Knowing which plasmids in this dataset carry *sul3* would be more useful than knowing which carry any sulphonamide resistance gene. The same is true for all other phenotype/genotype combinations.

1d) Reference 26 in this manuscript (PMID 28922058) delineates ColV plasmids according to their specific FII replicon types, and by the backbone insertion location of their antibiotic resistance regions. Plasmid lineages distinguished at replicon type level have distinguishable traits such as carriage of *Colla* determinants (see Table 2 in PMID 28922058), and resistance region locations can be used to further subdivide lineages. Reference 26 (Figure 3) and other work (PMID 32031906) shows that further mobile element-mediated evolution within acquired resistance regions can be characterised and traced. This prior knowledge must be considered here and should be referred to when attempting to subdivide plasmids. The relevance of ORF-based clustering relative to higher-resolution approaches for identifying and discriminating between plasmid lineages should be discussed.

2) Cointegrate identification and characterisation

While the description of cointegrates is an interesting part of this manuscript, it could be strengthened with consideration of the mechanisms responsible for cointegrate formation. It is not surprising that ColV plasmids have formed cointegrates, as IS26 has been reported in their acquired resistance regions (PMID 26855083, 28922058, 32031906). The role of IS26 in cointegrate formation has been well characterised (reviewed in PMID 38436262). This existing knowledge should be referred to when discussing cointegrate formation here and might be used to more accurately describe and explain the structures depicted in Figure 2A.

3) Plasmid co-carriage

I am not convinced that the data presented here is sufficient to declare that ColVLP carriage is strongly associated with carriage of these other plasmid types. Such a claim would require demonstrating that these plasmids are significantly underrepresented in *E. coli* genomes that do not contain ColVLP, and I expect that is not the case, particularly for I-complex plasmids, which are quite common in the species. Would the same co-carriage pattern be seen for pUT189-like plasmids (Ref 8 in this manuscript; PMID 39838323), or indeed for any other F-type plasmid?

4) Virulence gene recombination

This is a short part of the manuscript that needs further consideration and expansion. It appears that the data shown in Figure 4 is more complex than lines 281-283 and 415-417 suggest. First, genes of plasmid and chromosomal origin seem to be interspersed in multiple phylogenies, particularly *iroBCDEN*, *iutABCD/iutA*, and *sitABCD*. Second, the long branches that appear in regions of the phylogenies associated with plasmid-borne genes suggests that there is a large degree of sequence variation present. This might be the result of recombination. These genes should be examined, and if SNPs relative to the most common plasmid-borne alleles are clustered in patches, those patches should be compared to chromosomal alleles to determine whether they are the source of the recombinant parts.

Specific comments:

Throughout: Incompatibility (*Inc*) is a phenotype. Sequence-based approaches cannot predict complex compatibility reactions, so use of the term is best avoided. On line 254, it would be more accurate to say that replicon types are being investigated, not *Inc* groups.

Lines 73 and 138-141: Reference 16 from 2006 (PMID 16885466) is not very recent. Differences between ColV and ColBM plasmids have been known about for almost 20 years. Independent clustering of ColV and ColBM lineages is to be expected based on the existing literature. Considering that, should they be analysed together like this, or separately for greater resolution?

Lines 86-87: Backbone vs acquired AMR determinants

Acquired antibiotic resistance genes are not part of plasmid backbones. The convergence of virulence and resistance in these plasmids is explained by the insertion of mobile genetic elements and their associated resistance genes INTO the ColV backbone. Suggest modifying this sentence to reflect this, as stating that these determinants “exist on” the backbone is inaccurate and will lead to confusion.

Line 115: Recognising FII replicons

The replicons called FIC(FII) here are FII. FIC is a misleading entry in the PlasmidFinder database. Compare the structures of these to reference FII replicons and see prior studies on ColV plasmids that call them FII-18 or FII-24. The FAB formula should be presented in the format FII:FIA:FIB.

Line 119-120: Assuming conjugative ability

Conjugative ability should not be assumed from the presence of transfer genes. Even when all genes are present, insertions or polymorphisms in the transfer region might impact a plasmid’s ability to transfer. See PMID 26855083 for an example in a ColV plasmid.

Lines 458-451: Conclusions

This seems like too strong a statement to make here. The convergence of virulence and antibiotic resistance in the context of ColV plasmids has been demonstrated and detailed in previous studies and there do not seem to be any mechanistic explanations presented here.

Version 1:

Reviewer comments:

Reviewer #1

(Remarks to the Author)

I am satisfied that the revised manuscript and its clarifications / additional experiments are sound.

One thing that has confused me though is the language from Lines 318-323 presenting the new hybrid chromosomal/ColVLP sit operon data.

317 An alignment and comparison of non-
318 conserved sites revealed that these four ColVLP-associated sitABCD sequences were hybrids
319 between a ColVLP and chromosomal homolog, where the initial section (sitABC and start of
320 sitD) shared high conservation with the reference ColVLP-associated sequence, and the latter
321 section of sitD shared high conservation with the reference chromosomal sequence (Figure 4G).

Please double check that this statement and the data in Fig 4G are in agreement, as I'm interpreting the alignment it appears that the sitABC loci align to the chromosomal reference while sitD loci align with the plasmid reference...? If I'm miss reading the figure, my apologies.

Authors, thank you again for a pleasant, well written, and informative read.

Reviewer #3

(Remarks to the Author)

The manuscript has been improved by revision and most of my comments have been addressed by the changes made.

Notably, screening for IS26 has improved the description of cointegrates while further investigation of virulence gene alleles has revealed interesting examples of recombination between plasmid-borne and chromosomal genes.

I do not have any additional comments on this version of the manuscript.

I would like to commend the authors on this interesting study and their work to strengthen it in response to peer review.

REVIEWER COMMENTS

Reviewer #1 (Remarks to the Author):

The manuscript, "Convergence of plasmid-driven virulence and antibiotic resistance in *Escherichia coli*", by Lian et. al reports interesting findings on the composition of ColVLP plasmids, their diversity/similarity, co-occurrence with additional plasmids within top 100ST *E. coli* hosts, and the incidence of cointegration of ColVLP with some of these co-resident plasmids. The significance of the cointegration of ColVLP plasmids with additional plasmids leads to the potential for convergence of antibiotic resistance and virulence factors within the same plasmid and the study culminates in the evaluation of the virulence associated antimicrobial peptide activity/specificity of the unique plasmid encode protease variant OmpTp as compared to the chromosomally encoded variants OmpT1 and OmpT2 found in the ST95 representative strain MS7163. This is evaluated by assessing the in vitro proteolysis of potential substrates, which could be encountered in a human host, using a heterologous *E. coli* host and survival of MS7163, and various OmpT mutant combinations, to LL-37 (human cathelicidin). This study is well written and nicely outlines its findings with informative and well-organized figures, and I wish to let the authors know this was a pleasure to read. As a testament to this observation, I have very few critiques to provide and suggest them as potential points for improving clarity.

We thank the reviewer for the positive feedback and have addressed each of the points raised below.

Line 127 (Fig 1 legend) Specify "(imm)" abbreviation, presumably it's immunity

We have now specified this in the Figure legend (**new text underlined**).

Host isolate source, host species, replicon type, predicted conjugative ability, and the presence of colicins, colicin immunity genes (imm), virulence, and resistance features are annotated.

Line 170, is the third Cluster 1 plasmid, that does not encode *mcr-1*, so substantially different from the other hybrid plasmids that it could not be included in the alignment with the pMS7163A and pSTM6-275 references? Its conspicuous absence has puzzled me, and a line of justification for its exclusion would be helpful.

The third Cluster 1 plasmid, NZ_MG014722.1, is not substantially different from the remaining Cluster 1 plasmids. Pairwise comparisons to the two other *mcr-1* positive Cluster 1 plasmids indicate high homology. We have included the third plasmid in an **updated figure** (Figure 2A).

Figure 2. CoVLP co-integrates. (A) Pairwise comparisons of IncHI2/CoVLP co-integrate plasmids.

Reference sequences were a concatenation of the reference CoVLP pMS7136A (NZ_CP026854.1) and the reference IncHI2 plasmid pSTM6-275 (NZ_CP019647.1). Pairwise comparisons were performed using BLASTn and visualised using EasyFig (45) with a minimum alignment length of 1000bp. Features are coloured as follows: IncF *tra* transfer region – blue; CoVLP virulence loci – red; IncHI2 *tra1* transfer region – orange; IncHI2 *tra2* transfer region – green; IS26 – pink. **(B) Split decomposition network of CoVLP co-integrates.** Twenty non-redundant CoVLP co-integrates were identified from NCBI using the criteria of Liu C. M. et al. (6) and PlasmidFinder replicons (46). A network visualising plasmid relatedness was drawn using binary sequences generated using the ORF-based binarized structure network analyses tool. Clusters were determined based on a Dice index of >0.71. **(C) AMR genes (n) in CoVLP co-integrates and CoVLPs.** AMR genes were identified using ABRicate v0.8 against the ARG-ANNOT database (35) using a 100% query length threshold. Data is presented as a boxplot, where the box limits representing first and third quartiles, the internal line represents the median, and whiskers represent data within a ± 1.5 interquartile range. **** $p < 0.0001$ as measured using an unpaired, two-tailed t-test.

Lines 206 – 210, These three sentences are confusing as the IncHI2+ isolates are not all of human origin. As the information in the first sentence appears redundant and inaccurate, consider consolidating the three sentences on these lines, discussing IncHI2+ & IncN+, into two sentences.

The following changes have been made:

Original

Of these, plasmids in the IncHI2+ and IncN+ clusters were isolated from strains of human origin, and many carried the mcr-1 colistin resistance gene. Plasmids from the IncHI2+ cluster were isolated from human, avian, porcine, and fish isolates, indicative of zoonotic transmission and/or conjugation between human and animal isolates. All IncN+ CoVLPs were mcr-1 positive and from human isolates.

Changed

The five plasmids in the IncHI2+ cluster (of which two were *mcr-1* positive) were isolated from diverse sources including human, avian, porcine, and fish (Figure 2B), suggestive of zoonotic transmission and/or conjugative transfer between human and animal isolates. All IncN+ ColVLPs were *mcr-1* positive and from human isolates.

Line 228, referencing Fig 3A (rather than 3F) would be sufficient, as referring to Fig 3F this early distracts from the initial analysis being teed up for Fig 3A as it contains the 1469/9962 population data and 14.7%. Additionally, jumping to Fig3F here is tricky because 2/3 of the data in Fig3F is not explained until Line 263, so this leads to additional hunting and distracts from the very linear and organized story of the paper thus far.

We agree, and as suggested have changed the text so that we now refer to Fig 3A.

Fig 3B., it was difficult to find the ST types referenced in lines 232 – 234 as the organization of the plasmid occupancy % graphs for each ST type appears random (likely based on Top1-10 ST types). It would be easier to interpret and appreciate the dichotomous plasmid carriage of the ST types (ColVLPs alone vs. +additional plasmids) if they were organized with the upper row of graphs being of one type in numerical ST order and the lower row being the other type in numerical ST order. As it stands, it's hard to appreciate at a quick glance how the two populations are indeed unique from one another, and I found myself having to hunt for each ST type referenced (lines 232-234) among the 10 plots of Fig 3B.

We thank the reviewer for the comment and agree that Figure 3B can be changed to improve clarity. We have now done this, and organised the graphs based on the dichotomous plasmid carriage patterns as suggested.

Figure 3. ColVLPs and co-resident plasmids in a 100ST *E. coli* draft genome database. Presence of additional replicons in ColVLP+ draft genomes of the (A) 100ST database and (B) top 10 ColVLP+ STs within the 100ST database. The 100ST database consists of 9969 draft *E. coli* genomes from the top 100 STs in Enterobase. Replicon co-carryage network in the (C) 100ST, (D) ST95, and (E) ST131 (Clade B) databases. The ST95 and ST131 (Clade B) databases consist of 2118 and 344 draft genomes, respectively. ColVLPs were plotted as a separate node from IncF plasmids (IncF non-ColVLP). Total plasmid co-carryage events ($n \geq 10$) were plotted using Cytoscape v3.9.1. Red lines denote the top four associations with ColVLPs. Line thickness between nodes is proportional to the number of co-carryage events. (F) Percent of ColVLP+ isolates in the 100ST, ST95, and ST131 Clade B databases. ColVLP+ isolates were tallied regardless of co-carryage events. Plasmid combinations of ColVLP+ isolates in the

(G) 100ST, (H) ST95, and (I) ST131 (Clade B) databases. Plasmid combination frequency was normalised to the total number of ColVLP+ isolates in each database. The top six combinations of each database are shown. ColVLPs were identified using the criteria of Liu C. M. et al. (6). Additional replicons were identified using ABRicate v0.8 against the PlasmidFinder (40) database using a 100% query length threshold.

We have also changed the text as follows (new text underlined).

Original:

At the ST level, the distribution of additional replicons was ST-specific, with some STs primarily carrying ColVLPs alone (such as ST117, ST3580, ST95, ST73, ST162), but other STs largely carrying ColVLPs with >1 additional non-IncF replicon (such as ST428, ST23, ST88, ST57, ST602) (Figure 3B).

Changed:

At the ST level, the distribution of additional replicons was ST-specific, with >50% of strains in some STs carrying ColVLPs alone (ST73, ST95, ST117, ST162, ST3580), and other STs with >50% of strains carrying ColVLPs with one or more additional non-IncF replicon (ST23, ST57, ST88, ST428, ST602) (Figure 3B).

Fig 5H., Complementation of the ompT-null strain MS23578 (MS7163 Δ ompT2 Δ ompTp ompT1::Cm) with each of the individual ompT containing plasmids used for the proteolysis assays in MG1655 would be helpful in evaluating potential polar effects resulting from the introduction of these mutations. While the dose of the pOmpT encoded protease would likely exceed native levels, presumably the levels of each ompT variant on the pOmpT plasmid would be similar in this ompT-null strain and its inclusion would allow for survival-based contributions in their native host and pair nicely with the non-native MG1655 data presented in Fig 5G.

We agree this is a good idea and have complemented the ompT-null strain MS23578 with each individual OmpT plasmid, respectively. We tested the susceptibility of each complemented strain to LL-37, which revealed that plasmid complementation with any single OmpT variant did not significantly increase survival against LL-37.

This **new data** is appended to **Supplementary Figure 14 in a new panel B**. We have also revised the results in the manuscript to include these new data (**new text underlined**):

‘To investigate the contribution of the ColVLP-encoded OmpTp variant to LL-37 resistance, MS7163 *ompT* mutants expressing one, two or no active OmpT variants were generated using λ -Red mediated homologous recombination. The survival of each mutant was examined after 2 hours incubation with LL-37 in PBS. In these experiments, survival of the MS7163 triple mutant was significantly attenuated compared to the WT parent strain ($p < 0.0001$; Figure 5H), demonstrating that the combined absence of all three OmpT variants enhances susceptibility to LL-37. The expression of any single OmpT variant did not restore LL-37 resistance to WT levels (OmpTp [Figure 5H]; OmpT1 or OmpT2 [Supplementary Figure 14A]), even when expressed from a plasmid expression vector (Supplementary Figure 14B), revealing the requirement for a cooperative mode of resistance involving multiple OmpT variants. Despite this, the expression of OmpT1 and OmpTp consistently increased LL-37 resistance without always reaching statistical significance (Supplementary Figure 14A; Figure 5H). Notably, the co-expression of OmpTp+OmpT1 restored LL-37 resistance to a level comparable to WT (Figure 5H), but this was not observed for the co-expression of OmpTp+OmpT2 nor OmpT1+OmpT2 (Supplementary Figure 14A). Taken together, the data demonstrate a role for the ColVLP-encoded OmpTp variant in resistance to LL-37.’

Reviewer #2 (Remarks to the Author):

Review for Convergence of plasmid-driven virulence and antibiotic resistance in Escherichia coli

I enjoyed reading this one, the authors present on the genetic variation and possible gene exchange of virulence factors between ColV (and similar) plasmids.

Comments

* introduction is well described.

* The authors use a method of clustering plasmid sequences according to presence/absence of their expected gene list. This should work for the purposes used here, but I am curious why existing methods for plasmid taxonomy e.g. COPLA were not used. Admittedly, pMLST was used here. I usually see plasmid sequences compared with ANI or similar using the proportion of aligned sequence rather than the presence/absence approach used here. It could be that there is no real sequence similarity to make an alignment? Please clarify/justify this approach. See review of approaches in <https://www.sciencedirect.com/science/article/pii/S0147619X2300015X> or similar existing work.

We thank the reviewer for their insight and pointing us to COPLA. Our database does have sequence similarity to established PTUs and most ColVLPs (225/233) were assigned to, or closely related to, PTU-F_E. This new analysis is presented in a **new Supplementary Figure 1** (shown below). As our dataset consists of closely related plasmids with sufficiently detailed annotation, we could further improve clustering resolution, utilising our gene presence/absence methodology which is uniquely suited for such datasets with similar backbones. Indeed, doing so allowed us to separate our plasmids into several clusters with differing virulence and resistance gene carriage, which has implications for their contribution to pathogenesis when acquired by a host cell.

Supplementary Figure 1. Plasmid taxonomic unit (PTU) assignments of 233 ColVLPs generated using COPLA

We do acknowledge the following limitations of our methodology: (i) the methodology is annotation-dependent; (ii) independently acquired antibiotic resistance genes may influence clustering; and (iii) extremely distant plasmids will be clustered closer together than expected, due to many shared “0”s’ between pairwise alignments that actually indicate no homology. These caveats mean that this methodology is not suited to inferring plasmid evolution and should be avoided when clustering novel plasmids or a collection of distantly related plasmids, which is where an ANI-based tool such as COPLA is advantageous for determination of PTU assignment as an initial step.

We have added the COPLA assignment to the manuscript to indicate that most ColVLPs belong to PTU-F_E:

‘We generated a dataset comprising 233 non-redundant completely sequenced ColVLPs identified from PLSDDB (29) and NCBI RefSeq (30) according to previously described criteria (6). This stipulated that the ColVLPs contained >1 gene from at least four of the following sets: (i) *cvaABC* and *cvi*; (ii) *iroBCDEN*; (iii) *iucABCD* and *iutA*; (iv) *etsABC*; (v) *ompT* and *hlyF*; (vi) *sitABCD*. Plasmid taxonomic unit (PTU) classification based on average nucleotide identity revealed that most ColVLPs (225/233) belonged to, or were closely related to, PTU-F_E, conjugative F plasmids that reside in *E. coli* (31) (Supplementary Figure 1).’

We also agree that the limitations of our binary clustering methodology should be noted. Thus, we have included the following **new text in the discussion**.

‘We do note, however, that there are several limitations to our binary clustering methodology (36). The methodology is annotation-dependent, independently acquired antibiotic resistance genes may influence clustering, and non-related plasmids may be clustered closer together than expected due to many shared “0”s’ between pairwise alignments that indicate no homology. Given these limitations, we suggest this method is not optimal for inferring plasmid evolution, but instead is better suited to clustering annotated and related plasmids, such as ColVLPs.’

* as a corollary to that, i am curious which genes are informing those clusters as many of the virulence genes (fig 1 - blue) seem at odds with it. On the other hand, these plasmids look like mosaic structures with genes having different histories of evolution and exchange, and clustering on the presence/absence pattern on the whole might not mean very much.

The virulence loci represent a small proportion of all unique genes shared by in collection, and thus these genes only play a minor role in plasmid clustering. We recognise that plasmids are inherently mosaic structures due to frequent horizontal transmission and recombination. Hence, we are not using our plasmid clustering methodology to infer plasmid evolution. Rather, our methodology seeks to group ColVLPs according to similar gene content, independent of each plasmid’s evolutionary history. We have added the following **new text** in the discussion to clarify this:

‘Here, by utilising an ORF-based approach independent of genetic markers and a conserved backbone, we extend these existing analyses by providing a comprehensive, evolution-independent overview of currently sequenced ColVLPs (clustered by gene content). Our analysis identified diverse clusters of ColVLPs, providing a framework to characterise their contributions to ExPEC pathogenesis. We do note, however, that there are several limitations to our binary clustering methodology (36). The methodology is annotation-dependent, independently acquired antibiotic resistance genes may influence clustering, and non-related plasmids may be clustered closer together than expected due to many shared “0”s’ between pairwise alignments that indicate no homology. Given these limitations, we suggest this method is not optimal for inferring plasmid evolution, but instead is better suited to clustering annotated and related plasmids, such as ColVLPs.’

* Figure 1, perhaps a different colour coding for the different categories. Difficult to distinguish which column is host vs replicon vs conj. act. as they all use Red.

We have changed the colour scheme for the categories to improve visual clarity.

Figure 1. Cladogram of 233 ColVLPs. Unique ORFs from 233 ColVLPs were combined to form a hypothetical plasmid. Using sequence similarity searches against the hypothetical plasmid at an 80% nucleotide sequence identity and alignment length threshold, a binary sequence indicating ORF presence/absence for each plasmid was generated. Binary sequences were subjected to hierarchical clustering using manhattan distance and visualised as a cladogram. Plasmids were arranged into clusters ($n > 1$) based on the total within sum of square method, with minor ($n \leq 5$) and major clusters ($n > 5$). Host isolate source, host species, replicon type, predicted conjugative ability, and the presence of colicins, colicin immunity genes (imm), virulence, and resistance features were annotated. Feature prevalence is illustrated in a bar plot above the metadata. Potential conjugative ability was predicted based on the presence of all genes required for F-plasmid conjugation (33) determined using BLASTn at a 90% nucleotide sequence identity and alignment length threshold. Replicons were identified using ABRicate v0.8 against the PlasmidFinder database using a 100% query length threshold. Virulence genes were identified using a BLASTn search at a 90% nucleotide sequence identity and 95% alignment length threshold, with operons being considered present only if all genes were individually identified. AMR genes were identified using ABRicate v0.8 against the ARG-ANNOT database (34) using a 100% query length threshold and grouped based on antibiotic class, with the following abbreviations: AGly, aminoglycosides; Bla, beta-lactamases; Fcyn, fosfomycin; Flq, fluoroquinolones; MLS, macrolide-lincosamide-streptogramin; Phe, phenicols; Rif, rifampin; Sul, sulfonamides; Tet, tetracyclines; and Tmt, trimethoprim.

* Co integration result is interesting. Looks good.

Thank you

* Checking against other genomes of STs of interest will be easier in future with datasets such as "all the bacteria" <https://www.biorxiv.org/content/10.1101/2024.03.08.584059v3>

We agree. We are currently exploring ways to use this extremely valuable dataset in the future.

* Discussion : "Our study revealed that ColVLPs are largely restricted to E. coli, and only rarely found in *Klebsiella pneumoniae* or *Salmonella enterica*" I did not see such a result in Results. Perhaps I missed it.

Our search for ColVLPs was performed against the NCBI Refseq complete genomes database (02/01/2021), which contained the complete genomes of 820 *Klebsiella spp.*, 930 *Salmonella spp.*, and 1441 *Escherichia spp.* isolates. From this collection, 3, 6, and 224 ColVLPs were found, respectively, thus leading to our conclusion.

We have modified the relevant section (**new text underlined**) in the Materials and Methods to clarify this:

ColVLP-associated loci from the ColV plasmid pMS7163A (38) were used as a BLASTn query against the NCBI RefSeq (30) (21,034 chromosomes; 19,736 plasmids; 02/01/2021) and PLSDDB (29) (34,513 plasmids; 23/06/2021) databases to identify a non-redundant collection of 233 ColVLPs as described in Liu C. M. et al. (6). The NCBI RefSeq contained the complete genomes of 820 *Klebsiella spp.*, 930 *Salmonella spp.*, and 1441 *Escherichia spp.* isolates. Incomplete plasmids with ambiguous nucleotides were removed from further analyses. ColVLPs were assigned to pre-defined plasmid taxonomic units (PTUs) using COPLA (80), using both linear and circular topologies with default options.

* Methods (Bioinformatics - the part I know) look fine. Meets my standards.

Thank you

* You might get similar results with faster run times using USEARCH to replace BLAST and MAFFT to replace ClustalOmega.

We thank the reviewer for pointing this out and will explore these tools going forward.

Reviewer #3 (Remarks to the Author):

This manuscript describes the characterisation of a group of *E. coli* plasmids that carry both virulence and antibiotic resistance determinants. While the group of plasmids has been associated with virulence in *E. coli* that cause human and animal infections since the 1960s, examples reported over the last 20 years have acquired antibiotic resistance genes, representing a concerning convergence of clinically relevant phenotypes. The first half of this manuscript outlines an investigation of 233 complete plasmid sequences, and the second half describes the detailed characterisation of a plasmid-encoded outer membrane protease.

The characterisation of the outer membrane protease, OmpTp, is novel work that clearly defines its activity and has relevance to wider understanding of OmpT substrate specificity. The findings presented here suggest that OmpTp might contribute to *E. coli* virulence by conferring resistance to the human antimicrobial peptide cathelicidin (LL-37), which plays a protective antibacterial role in the urinary tract. This adds to the repertoire of well-characterised virulence determinants associated with this plasmid group, further strengthening their association with *E. coli* pathogenicity.

Sequence analysis includes: 1) clustering plasmids using an ORF-based approach and comparing cluster gene content, 2) screening plasmids for non-F-type replicons to explore cointegrate formation, 3) screening host genomes to describe co-resident plasmid types, and 4) comparing plasmid-associated virulence genes and their chromosomal homologues to assess the extent to which they recombine. This analysis has led to some interesting observations, particularly around the formation of cointegrates by members of this group. However, there are several issues with the analyses and conclusions drawn from them.

One issue significantly detracting from the analysis is the disconnect between it and the existing literature on this plasmid group. With the literature considered, it appears that some methods employed here are not suitable for obtaining insights that advance the understanding of these plasmids. For example, previous work has characterised the molecular events associated with the acquisition of AMR determinants, describing differences that delineate sub-lineages with much greater resolution than ORF-based clustering can provide. While the OmpTp work will be a valuable addition to the literature, the sequence-based analyses require reconsideration. I have provided more detailed comments below.

Comments on sequence analysis:

1) Plasmid clustering

The ORF-based clustering approach does not seem to be a stringent method for assessing plasmid relationships, particularly when complete plasmid sequences are available, as its simple presence/absence evaluation is blind to evolutionary events that shape and can be used to contextualise differences between plasmids. I have questions about the biological and evolutionary relevance of the clusters, and their overall pertinence given existing knowledge of ColV plasmid sub-lineages.

1a) Could a plasmid in cluster 1 or 2 be a deletion derivative of one in clusters 3-6? If so, a single molecular event might be having an overly significant impact on the shape of this cladogram, and clusters 1/2 might be artifacts.

We acknowledge there are limitations to our binary clustering methodology and addressed this in our response to reviewer 2. The methodology is annotation-dependent, and non-related plasmids may be clustered closer together than expected due to many shared "O"s' between pairwise alignments that indicate no homology. Despite this, we argue the method is most suited to clustering

related plasmids such as CoIVLPs, and we have added the limitations and our interpretation in the revised manuscript. To address the reviewer's concern, we note plasmids from clusters 1 and 2 do not form their unique clade/cluster based (solely) on deletion events, as they carry a relatively complete F transfer region and the majority of the CoIVLP-associated virulence factors. Rather, these plasmids form their own cluster due to the acquisition of additional open reading frames from IncHI2 plasmids.

To demonstrate this, we screened our 233 CoIVLP dataset using 150-bp bins from the IncHI2 plasmids as a query. This enabled clear visualisation of the acquisition of ~210kb and ~83kb of extra genetic material in plasmids from cluster 1 and cluster 2, respectively (see new Supplementary Figure 4). These DNA sequences are not found in plasmids of clusters 3-6. This supports our conclusion that plasmids from clusters 1 and 2 are different from the remaining CoIVLPs and warrant their unique position in the cladogram. In saying this, we also acknowledge that this could have arisen from a single molecular event (e.g. co-integration with co-resident plasmid), and we now acknowledge this in our revised manuscript (new text underlined text below).

'Outside of the major clusters, our analysis also identified minor clusters 1 and 2, which consisted of three plasmids each that carried additional IncHI2/IncHI2A and IncN replicons, respectively (Supplementary Figure 3). In addition to CoIVLP-associated virulence loci and numerous AMR genes, these hybrid plasmids also encoded *mcr-1* (Cluster 1: 2/3; Cluster 2: 3/3; Figure 1), conferring resistance to the last-line antibiotic colistin. A detailed analysis of these plasmids revealed two distinct lineages of CoIVLP hybrids that contain regions from IncHI2-like plasmids (Supplementary Figure 4), likely acquired from co-integration events.'

Supplementary Figure 4. Nucleotide identity (%) heatmap of 150-bp bins from the IncHI2 reference plasmid pSTM6-275 against the 233 CoIVLP cladogram. Bins were used in a BLASTn query against the 233 CoIVLPs at a 75% identity and length threshold.

1b) Lines 145-153: It appears that antibiotic resistance genes would have been included in the ORFs that informed this cladogram. If that is the case, are plasmids with large numbers of ARGs clustered together because they carry these ARGs? Do these clusters reflect evolutionary relationships, or have plasmids with convergent evolutionary trajectories (acquired the same genes in different events) been conflated?

Yes, the antibiotic resistance genes were included in the ORFs that informed the clustering. Despite this, we would not expect plasmids with similar ARGs to cluster together solely due to ARG acquisition, as each ORF is weighted equally. Rather, we expect backbone sequences (e.g. transfer, leading strand, replication-associated) to collectively contribute more to clustering. Our intention was not to infer plasmid evolutionary trajectory, but rather to cluster plasmids together based on similar gene content, independent of the method of acquisition.

We agree this is a limitation of our methodology and have added **new text** to discuss this in the revised manuscript (see below and response to reviewer 2).

‘We do note, however, that there are several limitations to our binary clustering methodology. The methodology is annotation-dependent, independently acquired antibiotic resistance genes may influence clustering, and non-related plasmids may be clustered closer together than expected due to many shared “0”s’ between pairwise alignments that indicate no homology. Given these limitations, we suggest this method is not optimal for inferring plasmid evolution, but instead is better suited to clustering annotated and related plasmids, such as ColVLPs.’

1c) Collapsing resistance gene data to phenotypic summaries is non-ideal when considering the evolution of these plasmids, as it masks data that is crucial when considering the convergence of virulence and resistance. For example, sulphonamide resistance can be conferred by *sul1*, *sul2*, or *sul3*, each of which has been mobilised independently and has a distinct evolutionary history. The rarer *sul3* has been reported in a ColV plasmid, with the mechanism for its acquisition described (see PMID 26855083). Knowing which plasmids in this dataset carry *sul3* would be more useful than knowing which carry any sulphonamide resistance gene. The same is true for all other phenotype/genotype combinations.

We agree with the reviewer that collapsing the genotypic data to a phenotypic summary can result in masking data that potentially informs the evolutionary trajectory of ColVLPs. The metadata featured on the cladogram was collapsed in this manner to provide an overview of the relevant phenotypes expected with the acquisition of a ColVLP. A more detailed genotypic summary is also included in the paper in Table S4. To specifically address this concern, we now provide a **new Supplementary Figure 2**, which visualises the presence/absence of each AMR gene determinant, including the *sul1-3* genes. We have also added new text (**underlined**) to describe this detail, using the *sul* genes (including *sul3*) as an example and added the reference suggested by the reviewer (PMID 26855083).

‘The prevalence of AMR genes between these major clusters also differed. ColVLPs from cluster 3 had low carriage rates of all AMR genes examined, while ColVLPs of cluster 6 had moderate rates of resistance towards aminoglycosides, beta-lactams, sulfonamides, tetracyclines, and trimethoprim antibiotics (~35-45%) (Figure 1). ColVLPs from clusters 4 and 5 possessed high AMR gene carriage across multiple antibiotic classes including aminoglycoside (~65% & ~100%), beta-lactam (~75% & ~75%), fluoroquinolone (~25% & ~45%), macrolide (~25% & ~50%), phenicol (~35% & ~70%), tetracycline (~75% & ~55%), and trimethoprim (~55% & ~90%) antibiotics (Figure 1). The presence/absence of each AMR gene determinant is detailed in Supplementary Data 4 and visualised

in Supplementary Figure 2, as the independent acquisition of each determinant can inform ColVLP evolutionary history, particularly in the case of rare determinants such as *sul3* (33).’

Supplementary Figure 2. AMR genes within the 233 ColVLPs. AMR genes were identified using ABRicate v0.8 against the ARG-ANNOT database (34) using a 100% query length threshold and plotted against the 233 ColVLP cladogram as described in Figure 1.

1d) Reference 26 in this manuscript (PMID 28922058) delineates ColV plasmids according to their specific FII replicon types, and by the backbone insertion location of their antibiotic resistance regions. Plasmid lineages distinguished at replicon type level have distinguishable traits such as carriage of *Colla* determinants (see Table 2 in PMID 28922058), and resistance region locations can be used to further subdivide lineages. Reference 26 (Figure 3) and other work (PMID 32031906) shows that further mobile element-mediated evolution within acquired resistance regions can be characterised and traced. This prior knowledge must be considered here and should be referred to when attempting to subdivide plasmids. The relevance of ORF-based clustering relative to higher-resolution approaches for identifying and discriminating between plasmid lineages should be discussed.

We agree and thank the reviewer for this insightful information. In the revised manuscript we discuss this work and cite the references indicated. We also present a **new Supplementary Figure 15**, which describes the molecular signatures utilised by references 32031906 and 28922058 to track FII-2/FIB-1 replicon-containing ColVLPs. This analysis demonstrates the FII-2/FIB-1 replicon-containing ColVLPs are present in a closely related group of cluster 6 plasmids.

The following **new text** has been added to the discussion, including the references suggested by the reviewer:

‘To date, most studies tracking ColVLP evolution have done so by tracing the acquisition and subsequent modification of mobile genetic elements carrying AMR determinants in a group of closely related FII-2/FIB-1-containing ColVLPs (26, 42). These molecular signatures include the presence of a class 1 integron truncated by IS26 (42), the acquisition of a Tn1721/Tn21 hybrid transposon, the acquisition and *in situ* modification of Tn1721, as well as the presence of Colicin Ia (26). While these approaches allow for high resolution understanding of plasmid evolution, they focus on sub-lineages of FII-2/FIB-1 ColVLPs (part of cluster 6; Supplementary Figure 15). Here, by utilising an ORF-based approach independent of genetic markers and a conserved backbone, we broaden these existing analyses by providing a comprehensive, evolution-independent overview of currently sequenced ColVLPs (clustered by gene content). Our analysis identified diverse clusters of ColVLPs, providing a framework to characterise their contributions to ExPEC pathogenesis.’

Supplementary Figure 15. Nucleotide identity (%) heatmap of (A) Class 1 integron and the IS26 truncation; (B) Plasmid pCERC4 Resistance Region 1; (C) Plasmid pCERC9/pCERC5 Resistance Region 2 against the 233 ColVLP cladogram. Heatmaps were generated using 50-bp bins from reference molecular signatures as a BLASTn query against the 233 ColVLPs, at 75% percent identity and alignment length thresholds. The IS26-truncated class 1 integron molecular signature was identified by (42), while pCERC4/pCERC9/pCERC5 resistance regions were identified by (26). The FII-2/FIB-1 plasmid pCERC4 is highlighted as reference.

2) Cointegrate identification and characterisation

While the description of cointegrates is an interesting part of this manuscript, it could be strengthened with consideration of the mechanisms responsible for cointegrate formation. It is not surprising that ColV plasmids have formed cointegrates, as IS26 has been reported in their acquired

resistance regions (PMID 26855083, 28922058, 32031906). The role of IS26 in cointegrate formation has been well characterised (reviewed in PMID 38436262). This existing knowledge should be referred to when discussing cointegrate formation here and might be used to more accurately describe and explain the structures depicted in Figure 2A.

We thank the reviewer for their insightful comments and have now mapped the IS26 elements into Figure 2A (see **updated figure** below). Indeed, we observed multiple copies of IS26 flanking the regions between the co-integrated ColVLP and co-resident plasmid. Although the sheer number of IS26 copies makes it difficult to visualise the precise co-integration site, this new information is consistent with the previously reported role of IS26 in co-integrate formation reported in the references noted by the reviewer.

Figure 2. ColVLP co-integrates. (A) Pairwise comparisons of IncHI2/ColVLP co-integrate plasmids. Reference sequences were a concatenation of the reference ColVLP pMS7163A (NZ_CP026854.1) and the reference IncHI2 plasmid pSTM6-275 (NZ_CP019647.1). Pairwise comparisons were performed using BLASTn and visualised using EasyFig (45) with a minimum alignment length of 1000bp. Features are coloured as follows: IncF tra transfer region – blue; ColVLP virulence loci – red; IncHI2 tra1 transfer region – orange; IncHI2 tra2 transfer region – green; IS26 – pink. **(B) Split decomposition network of ColVLP co-integrates.** Twenty non-redundant ColVLP co-integrates were identified from NCBI using the criteria of Liu C. M. et al. (6) and PlasmidFinder replicons (46). A network visualising plasmid relatedness was drawn using binary sequences generated using the ORF-based binarized structure network analyses tool. Clusters were determined based on a Dice index of >0.71. **(C) AMR genes (n) in ColVLP co-integrates and ColVLPs.** AMR genes were identified using ABRicate v0.8 against the ARG-ANNOT database (35) using a 100% query length threshold. Data is presented as a boxplot, where the box limits representing first and third quartiles, the internal line represents the median, and whiskers represent data within a ± 1.5 interquartile range. **** $p < 0.0001$ as measured using an unpaired, two-tailed t-test.

We have also modified the results by adding the following, including the references suggested by the reviewer:

'IS26, an insertion element associated with the mobilisation of resistance genes (41), has been implicated in co-integrate formation (42) and is known to be present in both ColVLPs (26, 33, 43) and IncHI2-like plasmids (39, 44). Investigation of our co-integrate plasmids for IS26 revealed that all co-integrates harboured multiple IS26 copies (between 6-15 copies) in regions between the ColVLP and co-resident plasmid (Figure 2A), consistent with previous reports that demonstrate the role of IS26 in co-integrate formation (41, 42, 45, 46).'

Furthermore, we investigated the 20 additional co-integrates for IS26, and found IS26 in 19/20 of these co-integrates. We have updated the pairwise comparisons in **Supplementary Figures 5-10** to include IS26 locations, and updated the results with the following:

'Interestingly, all co-integrates except for one IncI1⁺ plasmid contained IS26 insertion elements, mostly located in regions between the ColVLP and co-resident plasmid (Supplementary Figures 5-10), further supporting the role of IS26 in co-integrate formation.'

3) Plasmid co-carriage

I am not convinced that the data presented here is sufficient to declare that ColVLP carriage is strongly associated with carriage of these other plasmid types. Such a claim would require demonstrating that these plasmids are significantly underrepresented in E. coli genomes that do not contain ColVLP, and I expect that is not the case, particularly for I-complex plasmids, which are quite common in the species. Would the same co-carriage pattern be seen for pUTI89-like plasmids (Ref 8 in this manuscript; PMID 39838323), or indeed for any other F-type plasmid?

We agree with the reviewer that the co-carriage patterns described in Figure 3C would also be true with other F plasmids. We have therefore toned down the discussion section accordingly.

Previous:

"Indeed, the problem of incomplete plasmid assemblies is epitomized by our plasmid replicon co-carriage analyses, which indicate a strong association between ColVLP carriage and the IncI1, IncX, IncB/O/K/Z, and IncH replicons, but is unable to account for the co-occurrence of these replicons on the same backbone without closed plasmid assemblies."

Changed:

"Indeed, the problem of incomplete plasmid assemblies is a limitation of our plasmid replicon co-carriage analyses. Although our data show that ColVLP+ strains can carry additional replicons (IncI1, IncX, IncB/O/K/Z, and IncH as the most common), we cannot differentiate between independent co-occurrence or co-occurrence on the same plasmid backbone without closed assemblies."

The question referring to whether the same co-carriage pattern is seen for pUTI89-like plasmids (Ref 8 in this manuscript; PMID 39838323), or indeed for any other F-type plasmid is interesting. We considered this, but as our manuscript focusses on ColVLPs, we believe the analysis of co-carriage patterns targeting specific plasmids such as pUTI89-like plasmids is beyond the scope of the present study.

4) Virulence gene recombination

This is a short part of the manuscript that needs further consideration and expansion. It appears that the data shown in Figure 4 is more complex than lines 281-283 and 415-417 suggest. First, genes of plasmid and chromosomal origin seem to be interspersed in multiple phylogenies, particularly iroBCDEN, iutABCD/iutA, and sitABCD. Second, the long branches that appear in regions of the phylogenies associated with plasmid-borne genes suggests that there is a large degree of sequence variation present. This might be the result of recombination. These genes should be examined, and if

SNPs relative to the most common plasmid-borne alleles are clustered in patches, those patches should be compared to chromosomal alleles to determine whether they are the source of the recombinant parts.

We thank the reviewer for these additional insightful comments and have now investigated the phylogenies in greater detail following the suggestions provided. We manually checked every ColVLP-associated sequence which had a long branch and every sequence that did not cluster with their associated group.

For the virulence genes *iucABCD/iutA*, *shfF*, and *ompT*, all outliers were confirmed to be the results of inactivating mutations (frameshift or internal stop) or suspected mis-assembly. For *iss*, only one outlier was found, where a chromosomal-associated copy was found to cluster with ColVLP-associated homologs – while possibly recombination, more sequencing data is needed to confirm this observation.

For *iroBCDEN* and *sitABCD*, the ColVLP-associated homologs with long branches were also confirmed to contain inactivating mutations, and *sitABCD* additionally had two chromosomal homologs that clustered with ColVLP-associated copies due to suspected mis-assembly. Here, however, we were able to find evidence of recombination, where four instances of a ColVLP-associated *iroBCDEN* had replaced their chromosomal counterparts. We also found four *sitABCD* operons that were hybrids between a ColVLP-associated and chromosomal-associated sequence, and were able to narrow the recombination event to within the *sitD* gene.

In consideration of these new results, we added the following **new text** to the results section:

‘Here, we also noted rare instances where genes of ColVLP and chromosomal origin were interspersed in the phylogeny. To investigate these homologs for recombination, we manually checked the following for possible recombination: (i) every ColVLP-associated sequence which had a long branch indicative of large sequence variation; and (ii) every sequence that did not cluster with their associated group.

For the virulence genes *iucABCD/iutA*, *shfF*, and *ompT*, all outliers were confirmed to be associated with inactivating mutations (frameshift or internal stop) or suspected mis-assembly. For *iss*, only one outlier was found, where a chromosomal-associated copy clustered with ColVLP-associated homologs – while possibly recombination, more sequencing data is needed to confirm this interpretation.

For *iroBCDEN* and *sitABCD*, the ColVLP-associated homologs with long branches were also confirmed to contain inactivating mutations, and *sitABCD* additionally had two chromosomal homologs that clustered with ColVLP-associated copies due to suspected mis-assembly. Despite this, we were able to find convincing evidence of homologous recombination between ColVLPs and the chromosome. In the case of *iroBCDEN*, four fully intact operons that shared 99.97% nucleotide sequence similarity to a ColVLP-associated sequence were found on the *E. coli* chromosome (NZ_CP062985.1, NC_017632.1, NZ_CP012631.1, NZ_CP019777.1). More interestingly, with *sitABCD*, we found four ColVLP-associated sequences that formed their unique cluster within the chromosomal homologs. An alignment and comparison of non-conserved sites revealed that these four ColVLP-associated *sitABCD* sequences were hybrids between a ColVLP and chromosomal homolog, where the initial section (*sitABC* and start of *sitD*) shared high conservation with the reference ColVLP-associated sequence, and the latter section of *sitD* shared high conservation with the reference chromosomal sequence (Figure 4G).

Taken together, our results indicate that while most virulence loci carried on the ColVLP are distinct from their chromosomal counterparts, homologous recombination can occur, resulting in switching of the entire operon (as evidenced by *iroBCDEN*) or even a partial section of a virulence gene (as evidenced by *sitD*) between ColVLPs and the *E. coli* chromosome.'

We have also **modified Figure 4** by noting recombinant sequences and adding a panel to visualise the nucleotide identity of the non-conserved sites of the hybrid *sitABCD* genes compared to reference ColVLP- and chromosomal-associated sequences. **New text** added to the Figure Legend is underlined.

Figure 4. Phylogeny of the virulence genes (A) *iroBCDEN*, (B) *iucABCD/iutA*, (C) *shiF*, (D) *sitABCD*, (E) *iss*, and (F) *ompT*. Nucleotide sequences were extracted from the NCBI RefSeq (1,378 *E. coli* chromosomes) and 233 ColVLP collection, aligned using Clustal Omega, and visualised as a midpoint-rooted tree using FastTree and iTOL. Potential recombinant homologs are highlighted in blue. **(G) Nucleotide identity (%) comparing the non-conserved sites of the recombinant *sitABCD* operons**

with representative chromosomal and ColVLP homologs. Alignments between the four potential recombinant *sitABCD* operons (from NZ_CP019018.1, CP056868.1, NZ_CP032068.1, NZ_CP059908.1) and the representative homologs (ColVLP - NZ_CP026854.1, Chromosome - CP033884.1) were performed using CLC Main Workbench.

In addition, we have also **modified our discussion** to integrate these new findings:

Previous:

An intriguing aspect of our analysis was the distinct phylogenetic clustering of chromosomal- and ColVLP-encoded virulence genes, suggesting a lack of recombination between these regions (Figure 4). A limitation of this analysis was that the collection of chromosomal homologs was taken from completely sequenced *E. coli* genomes irrespective of ColVLP carriage and thus is likely predominantly from strains that do not carry ColVLPs. Indeed, a large collection of completely sequenced genomes belonging to ColVLP+ STs is lacking, making an assessment of recombination within these STs difficult. Thus, the full extent of recombination between chromosomal- and ColVLP-encoded virulence genes within these STs remains to be elucidated.

Changed:

An intriguing aspect of our analyses was the observation that some ColVLP-encoded virulence loci have recombined with their host chromosome, resulting in operon switching (*iroBCDEN*; Figure 4A) to hybrid sequences (*sitABCD*; Figure 4G). Here, our database consisted of completely sequenced *E. coli* genomes irrespective of ColVLP carriage and is likely predominantly from strains that do not normally carry ColVLPs. As recombination between the ColVLP and chromosome can only occur when a strain carries a ColVLP, we suspect these recombination events occur at a rate higher than what our results suggest. Elucidation of the full extent of recombination in ColVLP+ STs would require additional completely sequenced genomes.

Specific comments:

Throughout: Incompatibility (Inc) is a phenotype. Sequence-based approaches cannot predict complex compatibility reactions, so use of the term is best avoided. On line 254, it would be more accurate to say that replicon types are being investigated, not Inc groups.

We thank the reviewer for pointing this out and have amended the manuscript accordingly (**changes underlined**):

‘To investigate the replicons associated with ColVLP carriage, we plotted all significant co-carriage events in the 100ST database as a network. The top four replicons associated with ColVLP carriage were Inc1, IncX, IncB/O/K/Z, and IncH (Figure 3C). These co-carriage associations were also ST-dependent, as highlighted by the ColVLP-replicon associations of ST95 and ST131 (Clade B), two pandemic STs typically associated with ColVLP carriage, which were quantified with ST-specific databases of 2,118 and 344 draft genomes, respectively. In ColVLP+ ST95 (1268/2118; Figure 3F), the top four replicons associated with ColVLP carriage were (in order of magnitude): IncB/O/K/Z, IncX, IncH, and Inc1 (Figure 3D). In ColVLP+ ST131 Clade B genomes (191/344; Figure 3F), the top four replicons were: Inc1, IncH, IncX, and IncN (Figure 3E). Notably, strains of ST131 Clade B had higher replicon diversity compared to ST95. ST131 Clade B strains, in addition to carrying ColVLPs alone (31%), frequently contained combinations of ColVLP + Inc1 (24%); + Inc1 & IncH (10%); or + IncX (7%) (Figure 3I). In contrast, the majority of ST95 carried ColVLP alone (75%) or with an additional IncB/O/K/Z plasmid (13%) (Figure 3H). Taken together, the data demonstrates that ColVLP+ *E. coli*

strains frequently carry a diverse array of plasmid replicons, in part reflecting the existence of multiple co-resident plasmids in ST-specific patterns.'

Lines 73 and 138-141: Reference 16 from 2006 (PMID 16885466) is not very recent. Differences between ColV and ColBM plasmids have been known about for almost 20 years. Independent clustering of ColV and ColBM lineages is to be expected based on the existing literature. Considering that, should they be analysed together like this, or separately for greater resolution?

Our decision to analyse these plasmids together was based on the following: (i) ColBM plasmids still carry most of the virulence loci that define ColV plasmids, and thus have similar virulence characteristics; (ii) We could not be certain if a ColBM-negative plasmid had acquired ColBM and subsequently lost it, or if it never acquired ColBM in the first place.

Lines 86-87: Backbone vs acquired AMR determinants

Acquired antibiotic resistance genes are not part of plasmid backbones. The convergence of virulence and resistance in these plasmids is explained by the insertion of mobile genetic elements and their associated resistance genes INTO the ColV backbone. Suggest modifying this sentence to reflect this, as stating that these determinants "exist on" the backbone is inaccurate and will lead to confusion.

We have modified the sentence to reflect the comment as below (see **underlined changes**):

'Most importantly, several studies have reported increased antimicrobial resistance (AMR) among *E. coli* strains carrying ColVLPs (9, 25). The presence of AMR determinants within mobile genetic elements on ColVLPs represents a concerning convergence of virulence and resistance on a conjugative plasmid (26).'

Line 115: Recognising FII replicons

The replicons called FIC(FII) here are FII. FIC is a misleading entry in the PlasmidFinder database. Compare the structures of these to reference FII replicons and see prior studies on ColV plasmids that call them FII-18 or FII-24. The FAB formula should be presented in the format FII:FIA:FIB.

We thank the reviewer for bringing this to our attention. We have confirmed that FIC-4 is detecting the same gene as FII-18, and have removed FIC replicons from Figure 1, Supplementary Figure 3 and Supplementary Table S4. We have also changed the FAB formula to be presented in the appropriate format.

Line 119-120: Assuming conjugative ability

Conjugative ability should not be assumed from the presence of transfer genes. Even when all genes are present, insertions or polymorphisms in the transfer region might impact a plasmid's ability to transfer. See PMID 26855083 for an example in a ColV plasmid.

We agree, and we have amended the manuscript to include an appropriate level of caution (**changes are underlined**).

Figure 1 legend:

'Potential conjugative ability was predicted based on the presence of all genes required for F-plasmid conjugation (33) determined using BLASTn at a 90% nucleotide sequence identity and alignment length threshold, noting that a complete assessment of conjugation ability would require experimental validation.'

Results description:

More than half of the ColVLPs (52%) carried all genes required for F plasmid conjugation and are potentially conjugative (Figure 1), although transfer region polymorphisms and/or insertions may still impact transfer capability (33).

Lines 458-451: Conclusions

This seems like too strong a statement to make here. The convergence of virulence and antibiotic resistance in the context of ColV plasmids has been demonstrated and detailed in previous studies and there do not seem to be any mechanistic explanations presented here.

We have modified the conclusions statement as below (**new changes underlined**)

Previous:

'By demonstrating the carriage of multiple AMR genes on these plasmids, including co-integrate ColVLPs that harbour multiple replicons, our study describes a mechanism to explain plasmid-driven convergence of virulence and resistance in ExPEC.'

Changed:

'By demonstrating the carriage of multiple AMR genes on these plasmids, including co-integrate ColVLPs that harbour multiple replicons, our findings expand the role of ColVLPs in plasmid-driven convergence of virulence and resistance in ExPEC.

REVIEWER COMMENTS

Reviewer #1 (Remarks to the Author):

The manuscript, "Convergence of plasmid-driven virulence and antibiotic resistance in *Escherichia coli*", by Lian et. al reports interesting findings on the composition of ColVLP plasmids, their diversity/similarity, co-occurrence with additional plasmids within top 100ST *E. coli* hosts, and the incidence of cointegration of ColVLP with some of these co-resident plasmids. The significance of the cointegration of ColVP plasmids with additional plasmids leads to the potential for convergence of antibiotic resistance and virulence factors within the same plasmid and the study culminates in the evaluation of the virulence associated antimicrobial peptide activity/specificity of the unique plasmid encode protease variant OmpTp as compared to the chromosomally encoded variants OmpT1 and OmpT2 found in the ST95 representative strain MS7163. This is evaluated by assessing the in vitro proteolysis of potential substrates, which could be encountered in a human host, using a heterologous *E. coli* host and survival of MS7163, and various OmpT mutant combinations, to LL-37 (human cathelicidin). This study is well written and nicely outlines its findings with informative and well-organized figures, and I wish to let the authors know this was a pleasure to read. As a testament to this observation, I have very few critiques to provide and suggest them as potential points for improving clarity.

We thank the reviewer for the positive feedback and have addressed each of the points raised below.

Line 127 (Fig 1 legend) Specify "(imm)" abbreviation, presumably it's immunity

We have now specified this in the Figure legend (**new text underlined**).

Host isolate source, host species, replicon type, predicted conjugative ability, and the presence of colicins, colicin immunity genes (imm), virulence, and resistance features are annotated.

Line 170, is the third Cluster 1 plasmid, that does not encode *mcr-1*, so substantially different from the other hybrid plasmids that it could not be included in the alignment with the pMS7163A and pSTM6-275 references? Its conspicuous absence has puzzled me, and a line of justification for its exclusion would be helpful.

The third Cluster 1 plasmid, NZ_MG014722.1, is not substantially different from the remaining Cluster 1 plasmids. Pairwise comparisons to the two other *mcr-1* positive Cluster 1 plasmids indicate high homology. We have included the third plasmid in an **updated figure** (Figure 2A).

Figure 2. ColVLP co-integrates. (A) Pairwise comparisons of IncHI2/ColVLP co-integrate plasmids. Reference sequences were a concatenation of the reference ColVLP pMS7136A (NZ_CP026854.1) and the reference IncHI2 plasmid pSTM6-275 (NZ_CP019647.1). Pairwise comparisons were performed using BLASTn and visualised using EasyFig (45) with a minimum alignment length of 1000bp. Features are coloured as follows: IncF *tra* transfer region – blue; ColVLP virulence loci – red; IncHI2 *tra1* transfer region – orange; IncHI2 *tra2* transfer region – green; IS26 – pink. **(B) Split decomposition network of ColVLP co-integrates.** Twenty non-redundant ColVLP co-integrates were identified from NCBI using the criteria of Liu C. M. et al. (6) and PlasmidFinder replicons (46). A network visualising plasmid relatedness was drawn using binary sequences generated using the ORF-based binarized structure network analyses tool. Clusters were determined based on a Dice index of >0.71. **(C) AMR genes (n) in ColVLP co-integrates and ColVLPs.** AMR genes were identified using ABRicate v0.8 against the ARG-ANNOT database (35) using a 100% query length threshold. Data is presented as a boxplot, where the box limits representing first and third quartiles, the internal line represents the median, and whiskers represent data within a ± 1.5 interquartile range. **** $p < 0.0001$ as measured using an unpaired, two-tailed t-test.

Lines 206 – 210, These three sentences are confusing as the IncHI2+ isolates are not all of human origin. As the information in the first sentence appears redundant and inaccurate, consider consolidating the three sentences on these lines, discussing IncHI2+ & IncN+, into two sentences.

The following changes have been made:

Original

Of these, plasmids in the IncHI2+ and IncN+ clusters were isolated from strains of human origin, and many carried the *mcr-1* colistin resistance gene. Plasmids from the IncHI2+ cluster were isolated from human, avian, porcine, and fish isolates, indicative of zoonotic transmission and/or conjugation between human and animal isolates. All IncN+ ColVLPs were *mcr-1* positive and from human isolates.

Changed

The five plasmids in the IncHI2+ cluster (of which two were *mcr-1* positive) were isolated from human, avian, porcine, and fish sources, suggestive of zoonotic transmission and/or conjugation between human and animal isolates. All IncN+ ColVLPs were *mcr-1* positive and from human isolates.

Line 228, referencing Fig 3A (rather than 3F) would be sufficient, as referring to Fig 3F this early distracts from the initial analysis being teed up for Fig 3A as it contains the 1469/9962 population data and 14.7%. Additionally, jumping to Fig3F here is tricky because 2/3 of the data in Fig3F is not explained until Line 263, so this leads to additional hunting and distracts from the very linear and organized story of the paper thus far.

We agree, and as suggested have changed the text so that we now refer to Fig 3A.

Fig 3B., it was difficult to find the ST types referenced in lines 232 – 234 as the organization of the plasmid occupancy % graphs for each ST type appears random (likely based on Top1-10 ST types). It would be easier to interpret and appreciate the dichotomous plasmid carriage of the ST types (ColVLPs alone vs. +additional plasmids) if they were organized with the upper row of graphs being of one type in numerical ST order and the lower row being the other type in numerical ST order. As it stands, it's hard to appreciate at a quick glance how the two populations are indeed unique from one another, and I found myself having to hunt for each ST type referenced (lines 232-234) among the 10 plots of Fig 3B.

We thank the reviewer for the comment and agree that Figure 3B can be changed to improve clarity. We have now done this, and organised the graphs based on the dichotomous plasmid carriage patterns as suggested.

Figure 3. CoVLPs and co-resident plasmids in a 100ST *E. coli* draft genome database. Presence of additional replicons in CoVLP+ draft genomes of the (A) 100ST database and (B) top 10 CoVLP+ STs within the 100ST database. The 100ST database consists of 9969 draft *E. coli* genomes from the top 100 STs in Enterobase. Replicon co-carriage network in the (C) 100ST, (D) ST95, and (E) ST131 (Clade B) databases. The ST95 and ST131 (Clade B) databases consist of 2118 and 344 draft genomes, respectively. CoVLPs were plotted as a separate node from IncF plasmids (IncF non-CoVLP). Total plasmid co-carriage events ($n \geq 10$) were plotted using Cytoscape v3.9.1. Red lines denote the top four associations with CoVLPs. Line thickness between nodes is proportional to the number of co-carriage events. (F) Percent of CoVLP+ isolates in the 100ST, ST95, and ST131 Clade B databases. CoVLP+ isolates were tallied regardless of co-carriage events. Plasmid combinations of CoVLP+ isolates in the

(G) 100ST, (H) ST95, and (I) ST131 (Clade B) databases. Plasmid combination frequency was normalised to the total number of ColVLP+ isolates in each database. The top six combinations of each database are shown. ColVLPs were identified using the criteria of Liu C. M. et al. (6). Additional replicons were identified using ABRicate v0.8 against the PlasmidFinder (40) database using a 100% query length threshold.

We have also changed the text as follows (new text underlined).

Original:

At the ST level, the distribution of additional replicons was ST-specific, with some STs primarily carrying ColVLPs alone (such as ST117, ST3580, ST95, ST73, ST162), but other STs largely carrying ColVLPs with >1 additional non-IncF replicon (such as ST428, ST23, ST88, ST57, ST602) (Figure 3B).

Changed:

At the ST level, the distribution of additional replicons was ST-specific, with >50% of strains in some STs carrying ColVLPs alone (ST73, ST95, ST117, ST162, ST3580), and other STs with >50% of strains carrying ColVLPs with one or more additional non-IncF replicon (ST23, ST57, ST88, ST428, ST602) (Figure 3B).

Fig 5H., Complementation of the ompT-null strain MS23578 (MS7163 Δ ompT2 Δ ompTp ompT1::Cm) with each of the individual ompT containing plasmids used for the proteolysis assays in MG1655 would be helpful in evaluating potential polar effects resulting from the introduction of these mutations. While the dose of the pOmpT encoded protease would likely exceed native levels, presumably the levels of each ompT variant on the pOmpT plasmid would be similar in this ompT-null strain and its inclusion would allow for survival-based contributions in their native host and pair nicely with the non-native MG1655 data presented in Fig 5G.

We agree this is a good idea and have complemented the ompT-null strain MS23578 with each individual OmpT plasmid, respectively. We tested the susceptibility of each complemented strain to LL-37, which revealed that plasmid complementation with any single OmpT variant did not significantly increase survival against LL-37.

This **new data** is appended to **Supplementary Figure 14 in a new panel B**. We have also revised the results in the manuscript to include these new data (**new text underlined**):

‘To investigate the contribution of the ColVLP-encoded OmpTp variant to LL-37 resistance, MS7163 *ompT* mutants expressing one, two or no active OmpT variants were generated using λ -Red mediated homologous recombination. The survival of each mutant was examined after 2 hours incubation with LL-37 in PBS. In these experiments, survival of the MS7163 triple mutant was significantly attenuated compared to the WT parent strain ($p < 0.0001$; Figure 5H), demonstrating that the combined absence of all three OmpT variants enhances susceptibility to LL-37. The expression of any single OmpT variant did not restore LL-37 resistance to WT levels (OmpTp [Figure 5H]; OmpT1 or OmpT2 [Supplementary Figure 14A]), even when expressed from a plasmid expression vector (Supplementary Figure 14B), revealing the requirement for a cooperative mode of resistance involving multiple OmpT variants. Despite this, the expression of OmpT1 and OmpTp consistently increased LL-37 resistance without always reaching statistical significance (Supplementary Figure 14A; Figure 5H). Notably, the co-expression of OmpTp+OmpT1 restored LL-37 resistance to a level comparable to WT (Figure 5H), but this was not observed for the co-expression of OmpTp+OmpT2 nor OmpT1+OmpT2 (Supplementary Figure 14A). Taken together, the data demonstrate a role for the ColVLP-encoded OmpTp variant in resistance to LL-37.’

Reviewer #2 (Remarks to the Author):

Review for Convergence of plasmid-driven virulence and antibiotic resistance in Escherichia coli

I enjoyed reading this one, the authors present on the genetic variation and possible gene exchange of virulence factors between ColV (and similar) plasmids.

Comments

* introduction is well described.

* The authors use a method of clustering plasmid sequences according to presence/absence of their expected gene list. This should work for the purposes used here, but I am curious why existing methods for plasmid taxonomy e.g. COPLA were not used. Admittedly, pMLST was used here. I usually see plasmid sequences compared with ANI or similar using the proportion of aligned sequence rather than the presence/absence approach used here. It could be that there is no real sequence similarity to make an alignment? Please clarify/justify this approach. See review of approaches in <https://www.sciencedirect.com/science/article/pii/S0147619X2300015X> or similar existing work.

We thank the reviewer for their insight and pointing us to COPLA. Our database does have sequence similarity to established PTUs and most ColVLPs (225/233) were assigned to, or closely related to, PTU-F_E. This new analysis is presented in a **new Supplementary Figure 1** (shown below). As our dataset consists of closely related plasmids with sufficiently detailed annotation, we could further improve clustering resolution, utilising our gene presence/absence methodology which is uniquely suited for such datasets with similar backbones. Indeed, doing so allowed us to separate our plasmids into several clusters with differing virulence and resistance gene carriage, which has implications for their contribution to pathogenesis when acquired by a host cell.

Supplementary Figure 1. Plasmid taxonomic unit (PTU) assignments of 233 ColVLPs generated using COPLA

We do acknowledge the following limitations of our methodology: (i) the methodology is annotation-dependent; (ii) independently acquired antibiotic resistance genes may influence clustering; and (iii) extremely distant plasmids will be clustered closer together than expected, due to many shared “0”s’ between pairwise alignments that actually indicate no homology. These caveats mean that this methodology is not suited to inferring plasmid evolution and should be avoided when clustering novel plasmids or a collection of distantly related plasmids, which is where an ANI-based tool such as COPLA is advantageous for determination of PTU assignment as an initial step.

We have added the COPLA assignment to the manuscript to indicate that most ColVLPs belong to PTU-F_E:

‘We generated a dataset comprising 233 non-redundant completely sequenced ColVLPs identified from PLSDDB (29) and NCBI RefSeq (30) according to previously described criteria (6). This stipulated that the ColVLPs contained >1 gene from at least four of the following sets: (i) *cvaABC* and *cvi*; (ii) *iroBCDEN*; (iii) *iucABCD* and *iutA*; (iv) *etsABC*; (v) *ompT* and *hlyF*; (vi) *sitABCD*. Plasmid taxonomic unit (PTU) classification based on average nucleotide identity revealed that most ColVLPs (225/233) belonged to, or were closely related to, PTU-F_E, conjugative F plasmids that reside in *E. coli* (31) (Supplementary Figure 1).’

We also agree that the limitations of our binary clustering methodology should be noted. Thus, we have included the following **new text in the discussion**.

‘We do note, however, that there are several limitations to our binary clustering methodology (36). The methodology is annotation-dependent, independently acquired antibiotic resistance genes may influence clustering, and non-related plasmids may be clustered closer together than expected due to many shared “0”s’ between pairwise alignments that indicate no homology. Given these limitations, we suggest this method is not optimal for inferring plasmid evolution, but instead is better suited to clustering annotated and related plasmids, such as ColVLPs.’

* as a corollary to that, i am curious which genes are informing those clusters as many of the virulence genes (fig 1 - blue) seem at odds with it. On the other hand, these plasmids look like mosaic structures with genes having different histories of evolution and exchange, and clustering on the presence/absence pattern on the whole might not mean very much.

The virulence loci represent a small proportion of all unique genes shared by in collection, and thus these genes only play a minor role in plasmid clustering. We recognise that plasmids are inherently mosaic structures due to frequent horizontal transmission and recombination. Hence, we are not using our plasmid clustering methodology to infer plasmid evolution. Rather, our methodology seeks to group ColVLPs according to similar gene content, independent of each plasmid’s evolutionary history. We have added the following **new text** in the discussion to clarify this:

‘Here, by utilising an ORF-based approach independent of genetic markers and a conserved backbone, we broaden these existing analyses by providing a comprehensive, evolution-independent overview of currently sequenced ColVLPs (clustered by gene content). Our analysis identified diverse clusters of ColVLPs, providing a framework to characterise their contributions to ExPEC pathogenesis. We do note, however, that there are several limitations to our binary clustering methodology (36). The methodology is annotation-dependent, independently acquired antibiotic resistance genes may influence clustering, and non-related plasmids may be clustered closer together than expected due to many shared “0”s’ between pairwise alignments that indicate no homology. Given these limitations, we suggest this method is not optimal for inferring plasmid evolution, but instead is better suited to clustering annotated and related plasmids, such as ColVLPs.’

* Figure 1, perhaps a different colour coding for the different categories. Difficult to distinguish which column is host vs replicon vs conj. act. as they all use Red.

We have changed the colour scheme for the categories to improve visual clarity.

Figure 1. Cladogram of 233 ColVLPs. Unique ORFs from 233 ColVLPs were combined to form a hypothetical plasmid. Using sequence similarity searches against the hypothetical plasmid at an 80% nucleotide sequence identity and alignment length threshold, a binary sequence indicating ORF presence/absence for each plasmid was generated. Binary sequences were subjected to hierarchical clustering using manhattan distance and visualised as a cladogram. Plasmids were arranged into clusters ($n > 1$) based on the total within sum of square method, with minor ($n \leq 5$) and major clusters ($n > 5$). Host isolate source, host species, replicon type, predicted conjugative ability, and the presence of colicins, colicin immunity genes (*imm*), virulence, and resistance features were annotated. Feature prevalence is illustrated in a bar plot above the metadata. Potential conjugative ability was predicted based on the presence of all genes required for F-plasmid conjugation (33) determined using BLASTn at a 90% nucleotide sequence identity and alignment length threshold. Replicons were identified using ABRicate v0.8 against the PlasmidFinder database using a 100% query length threshold. Virulence genes were identified using a BLASTn search at a 90% nucleotide sequence identity and 95% alignment length threshold, with operons being considered present only if all genes were individually identified. AMR genes were identified using ABRicate v0.8 against the ARG-ANNOT database (34) using a 100% query length threshold and grouped based on antibiotic class, with the following abbreviations: AGly, aminoglycosides; Bla, beta-lactamases; Fcyn, fosfomycin; Flq, fluoroquinolones; MLS, macrolide-lincosamide-streptogramin; Phe, phenicols; Rif, rifampin; Sul, sulfonamides; Tet, tetracyclines; and Tmt, trimethoprim.

* Co integration result is interesting. Looks good.

Thank you

* Checking against other genomes of STs of interest will be easier in future with datasets such as "all the bacteria" <https://www.biorxiv.org/content/10.1101/2024.03.08.584059v3>

We agree. We are currently exploring ways to use this extremely valuable dataset in the future.

* Discussion : "Our study revealed that ColVLPs are largely restricted to E. coli, and only rarely found in Klebsiella pneumoniae or Salmonella enterica" I did not see such a result in Results. Perhaps I missed it.

Our search for ColVLPs was performed against the NCBI Refseq complete genomes database (02/01/2021), which contained the complete genomes of 820 *Klebsiella spp.*, 930 *Salmonella spp.*, and 1441 *Escherichia spp.* isolates. From this collection, 3, 6, and 224 ColVLPs were found, respectively, thus leading to our conclusion.

We have modified the relevant section (**new text underlined**) in the Materials and Methods to clarify this:

ColVLP-associated loci from the ColV plasmid pMS7163A (38) were used as a BLASTn query against the NCBI RefSeq (30) (21,034 chromosomes; 19,736 plasmids; 02/01/2021) and PLSDb (29) (34,513 plasmids; 23/06/2021) databases to identify a non-redundant collection of 233 ColVLPs as described in Liu C. M. et al. (6). The NCBI RefSeq contained the complete genomes of 820 *Klebsiella spp.*, 930 *Salmonella spp.*, and 1441 *Escherichia spp.* isolates. Incomplete plasmids with ambiguous nucleotides were removed from further analyses. ColVLPs were assigned to pre-defined plasmid taxonomic units (PTUs) using COPLA (80), using both linear and circular topologies with default options.

* Methods (Bioinformatics - the part I know) look fine. Meets my standards.

Thank you

* You might get similar results with faster run times using USEARCH to replace BLAST and MAFFT to replace ClustalOmega.

We thank the reviewer for pointing this out and will explore these tools going forward.

Reviewer #3 (Remarks to the Author):

This manuscript describes the characterisation of a group of *E. coli* plasmids that carry both virulence and antibiotic resistance determinants. While the group of plasmids has been associated with virulence in *E. coli* that cause human and animal infections since the 1960s, examples reported over the last 20 years have acquired antibiotic resistance genes, representing a concerning convergence of clinically relevant phenotypes. The first half of this manuscript outlines an investigation of 233 complete plasmid sequences, and the second half describes the detailed characterisation of a plasmid-encoded outer membrane protease.

The characterisation of the outer membrane protease, OmpTp, is novel work that clearly defines its activity and has relevance to wider understanding of OmpT substrate specificity. The findings presented here suggest that OmpTp might contribute to *E. coli* virulence by conferring resistance to the human antimicrobial peptide cathelicidin (LL-37), which plays a protective antibacterial role in the urinary tract. This adds to the repertoire of well-characterised virulence determinants associated with this plasmid group, further strengthening their association with *E. coli* pathogenicity.

Sequence analysis includes: 1) clustering plasmids using an ORF-based approach and comparing cluster gene content, 2) screening plasmids for non-F-type replicons to explore cointegrate formation, 3) screening host genomes to describe co-resident plasmid types, and 4) comparing plasmid-associated virulence genes and their chromosomal homologues to assess the extent to which they recombine. This analysis has led to some interesting observations, particularly around the formation of cointegrates by members of this group. However, there are several issues with the analyses and conclusions drawn from them.

One issue significantly detracting from the analysis is the disconnect between it and the existing literature on this plasmid group. With the literature considered, it appears that some methods employed here are not suitable for obtaining insights that advance the understanding of these plasmids. For example, previous work has characterised the molecular events associated with the acquisition of AMR determinants, describing differences that delineate sub-lineages with much greater resolution than ORF-based clustering can provide. While the OmpTp work will be a valuable addition to the literature, the sequence-based analyses require reconsideration. I have provided more detailed comments below.

Comments on sequence analysis:

1) Plasmid clustering

The ORF-based clustering approach does not seem to be a stringent method for assessing plasmid relationships, particularly when complete plasmid sequences are available, as its simple presence/absence evaluation is blind to evolutionary events that shape and can be used to contextualise differences between plasmids. I have questions about the biological and evolutionary relevance of the clusters, and their overall pertinence given existing knowledge of ColV plasmid sub-lineages.

1a) Could a plasmid in cluster 1 or 2 be a deletion derivative of one in clusters 3-6? If so, a single molecular event might be having an overly significant impact on the shape of this cladogram, and clusters 1/2 might be artifacts.

We acknowledge there are limitations to our binary clustering methodology and addressed this in our response to reviewer 2. The methodology is annotation-dependent, and non-related plasmids may be clustered closer together than expected due to many shared "0"s' between pairwise alignments that indicate no homology. Despite this, we argue the method is most suited to clustering

related plasmids such as ColVLPs, and we have added the limitations and our interpretation in the revised manuscript. To address the reviewer’s concern, we note plasmids from clusters 1 and 2 do not form their unique clade/cluster based (solely) on deletion events, as they carry a relatively complete F transfer region and the majority of the ColVLP-associated virulence factors. Rather, these plasmids form their own cluster due to the acquisition of additional open reading frames from IncHI2 plasmids.

To demonstrate this, we screened our 233 ColVLP dataset using 150-bp bins from the IncHI2 plasmids as a query. This enabled clear visualisation of the acquisition of ~210kb and ~83kb of extra genetic material in plasmids from cluster 1 and cluster 2, respectively (see **new Supplementary Figure 4**). These DNA sequences are not found in plasmids of clusters 3-6. This supports our conclusion that plasmids from clusters 1 and 2 are different from the remaining ColVLPs and warrant their unique position in the cladogram. In saying this, we also acknowledge that this could have arisen from a single molecular event (e.g. co-integration with co-resident plasmid), and we now acknowledge this in our revised manuscript (**new text underlined text below**).

‘Outside of the major clusters, our analysis also identified minor clusters 1 and 2, which consisted of three plasmids each that carried additional IncHI2/IncHI2A and IncN replicons, respectively (Supplementary Figure 3). In addition to ColVLP-associated virulence loci and numerous AMR genes, these hybrid plasmids also encoded *mcr-1* (Cluster 1: 2/3; Cluster 2: 3/3; Figure 1), conferring resistance to the last-line antibiotic colistin. A detailed analysis of these plasmids revealed two distinct lineages of ColVLP hybrids that contain regions from IncHI2-like plasmids (Supplementary Figure 4), likely acquired from co-integration events.’

Supplementary Figure 4. Nucleotide identity (%) heatmap of 150-bp bins from the IncHI2 reference plasmid pSTM6-275 against the 233 ColVLP cladogram. Bins were used in a BLASTn query against the 233 ColVLPs at a 75% identity and length threshold.

1b) Lines 145-153: It appears that antibiotic resistance genes would have been included in the ORFs that informed this cladogram. If that is the case, are plasmids with large numbers of ARGs clustered together because they carry these ARGs? Do these clusters reflect evolutionary relationships, or have plasmids with convergent evolutionary trajectories (acquired the same genes in different events) been conflated?

Yes, the antibiotic resistance genes were included in the ORFs that informed the clustering. Despite this, we would not expect plasmids with similar ARGs to cluster together solely due to ARG acquisition, as each ORF is weighted equally. Rather, we expect backbone sequences (e.g. transfer, leading strand, replication-associated) to collectively contribute more to clustering. Our intention was not to infer plasmid evolutionary trajectory, but rather to cluster plasmids together based on similar gene content, independent of the method of acquisition.

We agree this is a limitation of our methodology and have added **new text** to discuss this in the revised manuscript (see below and response to reviewer 2).

‘We do note, however, that there are several limitations to our binary clustering methodology. The methodology is annotation-dependent, independently acquired antibiotic resistance genes may influence clustering, and non-related plasmids may be clustered closer together than expected due to many shared “0”s’ between pairwise alignments that indicate no homology. Given these limitations, we suggest this method is not optimal for inferring plasmid evolution, but instead is better suited to clustering annotated and related plasmids, such as CoVLPs.’

1c) Collapsing resistance gene data to phenotypic summaries is non-ideal when considering the evolution of these plasmids, as it masks data that is crucial when considering the convergence of virulence and resistance. For example, sulphonamide resistance can be conferred by *sul1*, *sul2*, or *sul3*, each of which has been mobilised independently and has a distinct evolutionary history. The rarer *sul3* has been reported in a CoIV plasmid, with the mechanism for its acquisition described (see PMID 26855083). Knowing which plasmids in this dataset carry *sul3* would be more useful than knowing which carry any sulphonamide resistance gene. The same is true for all other phenotype/genotype combinations.

We agree with the reviewer that collapsing the genotypic data to a phenotypic summary can result in masking data that potentially informs the evolutionary trajectory of CoVLPs. The metadata featured on the cladogram was collapsed in this manner to provide an overview of the relevant phenotypes expected with the acquisition of a CoVLP. A more detailed genotypic summary is also included in the paper in Table S4. To specifically address this concern, we now provide a **new Supplementary Figure 2**, which visualises the presence/absence of each AMR gene determinant, including the *sul1-3* genes. We have also added new text (**underlined**) to describe this detail, using the *sul* genes (including *sul3*) as an example and added the reference suggested by the reviewer (PMID 26855083).

‘The prevalence of AMR genes between these major clusters also differed. CoVLPs from cluster 3 had low carriage rates of all AMR genes examined, while CoVLPs of cluster 6 had moderate rates of resistance towards aminoglycosides, beta-lactams, sulfonamides, tetracyclines, and trimethoprim antibiotics (~35-45%) (Figure 1). CoVLPs from clusters 4 and 5 possessed high AMR gene carriage across multiple antibiotic classes including aminoglycoside (~65% & ~100%), beta-lactam (~75% & ~75%), fluoroquinolone (~25% & ~45%), macrolide (~25% & ~50%), phenicol (~35% & ~70%), tetracycline (~75% & ~55%), and trimethoprim (~55% & ~90%) antibiotics (Figure 1). The presence/absence of each AMR gene determinant is detailed in Supplementary Data 4 and visualised

in Supplementary Figure 2, as the independent acquisition of each determinant can inform CoVLP evolutionary history, particularly in the case of rare determinants such as *sul3* (33).'

Supplementary Figure 2. AMR genes within the 233 CoVLPs. AMR genes were identified using ABRicate v0.8 against the ARG-ANNOT database (34) using a 100% query length threshold and plotted against the 233 CoVLP cladogram as described in Figure 1.

1d) Reference 26 in this manuscript (PMID 28922058) delineates CoV plasmids according to their specific FII replicon types, and by the backbone insertion location of their antibiotic resistance regions. Plasmid lineages distinguished at replicon type level have distinguishable traits such as carriage of *Colla* determinants (see Table 2 in PMID 28922058), and resistance region locations can be used to further subdivide lineages. Reference 26 (Figure 3) and other work (PMID 32031906) shows that further mobile element-mediated evolution within acquired resistance regions can be characterised and traced. This prior knowledge must be considered here and should be referred to when attempting to subdivide plasmids. The relevance of ORF-based clustering relative to higher-resolution approaches for identifying and discriminating between plasmid lineages should be discussed.

We agree and thank the reviewer for this insightful information. In the revised manuscript we discuss this work and cite the references indicated. We also present a **new Supplementary Figure 15**, which describes the molecular signatures utilised by references 32031906 and 28922058 to track FII-2/FIB-1 replicon-containing CoVLPs. This analysis demonstrates the FII-2/FIB-1 replicon-containing CoVLPs are present in a closely related group of cluster 6 plasmids.

The following **new text** has been added to the discussion, including the references suggested by the reviewer:

‘To date, most studies tracking ColVLP evolution have done so by tracing the acquisition and subsequent modification of mobile genetic elements carrying AMR determinants in a group of closely related FII-2/FIB-1-containing ColVLPs (26, 42). These molecular signatures include the presence of a class 1 integron truncated by IS26 (42), the acquisition of a Tn1721/Tn21 hybrid transposon, the acquisition and *in situ* modification of Tn1721, as well as the presence of Colicin Ia (26). While these approaches allow for high resolution understanding of plasmid evolution, they focus on sub-lineages of FII-2/FIB-1 ColVLPs (part of cluster 6; Supplementary Figure 15). Here, by utilising an ORF-based approach independent of genetic markers and a conserved backbone, we broaden these existing analyses by providing a comprehensive, evolution-independent overview of currently sequenced ColVLPs (clustered by gene content). Our analysis identified diverse clusters of ColVLPs, providing a framework to characterise their contributions to ExPEC pathogenesis.’

Supplementary Figure 15. Nucleotide identity (%) heatmap of (A) Class 1 integron and the IS26 truncation; (B) Plasmid pCERC4 Resistance Region 1; (C) Plasmid pCERC9/pCERC5 Resistance Region 2 against the 233 ColVLP cladogram. Heatmaps were generated using 50-bp bins from reference molecular signatures as a BLASTn query against the 233 ColVLPs, at 75% percent identity and alignment length thresholds. The IS26-truncated class 1 integron molecular signature was identified by (42), while pCERC4/pCERC9/pCERC5 resistance regions were identified by (26). The FII-2/FIB-1 plasmid pCERC4 is highlighted as reference.

2) Cointegrate identification and characterisation

While the description of cointegrates is an interesting part of this manuscript, it could be strengthened with consideration of the mechanisms responsible for cointegrate formation. It is not surprising that ColV plasmids have formed cointegrates, as IS26 has been reported in their acquired

resistance regions (PMID 26855083, 28922058, 32031906). The role of IS26 in cointegrate formation has been well characterised (reviewed in PMID 38436262). This existing knowledge should be referred to when discussing cointegrate formation here and might be used to more accurately describe and explain the structures depicted in Figure 2A.

We thank the reviewer for their insightful comments and have now mapped the IS26 elements into Figure 2A (see **updated figure** below). Indeed, we observed multiple copies of IS26 flanking the regions between the co-integrated ColVLP and co-resident plasmid. Although the sheer number of IS26 copies makes it difficult to visualise the precise co-integration site, this new information is consistent with the previously reported role of IS26 in co-integrate formation reported in the references noted by the reviewer.

Figure 2. ColVLP co-integrates. (A) Pairwise comparisons of IncHI2/ColVLP co-integrate plasmids. Reference sequences were a concatenation of the reference ColVLP pMS7163A (NZ_CP026854.1) and the reference IncHI2 plasmid pSTM6-275 (NZ_CP019647.1). Pairwise comparisons were performed using BLASTn and visualised using EasyFig (45) with a minimum alignment length of 1000bp. Features are coloured as follows: IncF *tra* transfer region – blue; ColVLP virulence loci – red; IncHI2 *tra1* transfer region – orange; IncHI2 *tra2* transfer region – green; IS26 – pink. **(B) Split decomposition network of ColVLP co-integrates.** Twenty non-redundant ColVLP co-integrates were identified from NCBI using the criteria of Liu C. M. et al. (6) and PlasmidFinder replicons (46). A network visualising plasmid relatedness was drawn using binary sequences generated using the ORF-based binarized structure network analyses tool. Clusters were determined based on a Dice index of >0.71. **(C) AMR genes (n) in ColVLP co-integrates and ColVLPs.** AMR genes were identified using ABRicate v0.8 against the ARG-ANNOT database (35) using a 100% query length threshold. Data is presented as a boxplot, where the box limits representing first and third quartiles, the internal line represents the median, and whiskers represent data within a ± 1.5 interquartile range. **** $p < 0.0001$ as measured using an unpaired, two-tailed t-test.

We have also modified the results by adding the following, including the references suggested by the reviewer:

'IS26, an insertion element associated with the mobilisation of resistance genes (41), has been previously implicated in co-integrate formation (42) and is known to be present in both ColVLPs (26, 33, 43) and IncHI2-like plasmids (39, 44). Investigation of our co-integrate plasmids for IS26 revealed that all co-integrates harboured multiple IS26 copies (between 6-15 copies) in regions between the ColVLP and co-resident plasmid (Figure 2A), consistent with previous reports that demonstrate the role of IS26 in co-integrate formation (41, 42, 45, 46).'

Furthermore, we investigated the 20 additional co-integrates for IS26, and found IS26 in 19/20 of these co-integrates. We have updated the pairwise comparisons in **Supplementary Figures 5-10** to include IS26 locations, and updated the results with the following:

'Interestingly, all co-integrates except for one IncI1⁺ plasmid contained IS26 insertion elements, mostly located in regions between the ColVLP and co-resident plasmid (Supplementary Figures 5-10), further supporting the role of IS26 in co-integrate formation.'

3) Plasmid co-carriage

I am not convinced that the data presented here is sufficient to declare that ColVLP carriage is strongly associated with carriage of these other plasmid types. Such a claim would require demonstrating that these plasmids are significantly underrepresented in *E. coli* genomes that do not contain ColVLP, and I expect that is not the case, particularly for I-complex plasmids, which are quite common in the species. Would the same co-carriage pattern be seen for pUTI89-like plasmids (Ref 8 in this manuscript; PMID 39838323), or indeed for any other F-type plasmid?

We agree with the reviewer that the co-carriage patterns described in Figure 3C would also be true with other F plasmids. We have therefore toned down the discussion section accordingly.

Previous:

"Indeed, the problem of incomplete plasmid assemblies is epitomized by our plasmid replicon co-carriage analyses, which indicate a strong association between ColVLP carriage and the IncI1, IncX, IncB/O/K/Z, and IncH replicons, but is unable to account for the co-occurrence of these replicons on the same backbone without closed plasmid assemblies."

Changed:

"Indeed, the problem of incomplete plasmid assemblies is a limitation of our plasmid replicon co-carriage analyses. Although our data show that ColVLP+ strains can carry additional replicons (IncI1, IncX, IncB/O/K/Z, and IncH as the most common), we cannot differentiate between independent co-occurrence or co-occurrence on the same plasmid backbone without closed assemblies."

The question referring to whether the same co-carriage pattern is seen for pUTI89-like plasmids (Ref 8 in this manuscript; PMID 39838323), or indeed for any other F-type plasmid is interesting. We considered this, but as our manuscript focusses on ColVLPs, we believe the analysis of co-carriage patterns targeting specific plasmids such as pUTI89-like plasmids is beyond the scope of the present study.

4) Virulence gene recombination

This is a short part of the manuscript that needs further consideration and expansion. It appears that the data shown in Figure 4 is more complex than lines 281-283 and 415-417 suggest. First, genes of plasmid and chromosomal origin seem to be interspersed in multiple phylogenies, particularly iroBCDEN, iutABCD/iutA, and sitABCD. Second, the long branches that appear in regions of the phylogenies associated with plasmid-borne genes suggests that there is a large degree of sequence variation present. This might be the result of recombination. These genes should be examined, and if

SNPs relative to the most common plasmid-borne alleles are clustered in patches, those patches should be compared to chromosomal alleles to determine whether they are the source of the recombinant parts.

We thank the reviewer for these additional insightful comments and have now investigated the phylogenies in greater detail following the suggestions provided. We manually checked every ColVLP-associated sequence which had a long branch and every sequence that did not cluster with their associated group.

For the virulence genes *iucABCD/iutA*, *shiF*, and *ompT*, all outliers were confirmed to be the results of inactivating mutations (frameshift or internal stop) or suspected mis-assembly. For *iss*, only one outlier was found, where a chromosomal-associated copy was found to cluster with ColVLP-associated homologs – while possibly recombination, more sequencing data is needed to confirm this observation.

For *iroBCDEN* and *sitABCD*, the ColVLP-associated homologs with long branches were also confirmed to contain inactivating mutations, and *sitABCD* additionally had two chromosomal homologs that clustered with ColVLP-associated copies due to suspected mis-assembly. Here, however, we were able to find evidence of recombination, where four instances of a ColVLP-associated *iroBCDEN* had replaced their chromosomal counterparts. We also found four *sitABCD* operons that were hybrids between a ColVLP-associated and chromosomal-associated sequence, and were able to narrow the recombination event to within the *sitD* gene.

In consideration of these new results, we added the following **new text** to the results section:

‘Here, we also noted rare instances where genes of ColVLP and chromosomal origin were interspersed in the phylogeny. To investigate these homologs for recombination, we manually checked the following for possible recombination: (i) every ColVLP-associated sequence which had a long branch indicative of large sequence variation; and (ii) every sequence that did not cluster with their associated group.

For the virulence genes *iucABCD/iutA*, *shiF*, and *ompT*, all outliers were confirmed to be the results of inactivating mutations (frameshift or internal stop) or suspected mis-assembly. For *iss*, only one outlier was found, where a chromosomal-associated copy was found to cluster with ColVLP-associated homologs – while possibly recombination, more sequencing data is needed to confirm this observation.

For *iroBCDEN* and *sitABCD*, the ColVLP-associated homologs with long branches were also confirmed to contain inactivating mutations, and *sitABCD* additionally had two chromosomal homologs that clustered with ColVLP-associated copies due to suspected mis-assembly. Despite this, we were able to find convincing evidence of homologous recombination between ColVLPs and the chromosome. In the case of *iroBCDEN*, four fully intact operons that shared 99.97% nucleotide sequence similarity to a ColVLP-associated sequence were found on the *E. coli* chromosome (NZ_CP062985.1, NC_017632.1, NZ_CP012631.1, NZ_CP019777.1). More interestingly, with *sitABCD*, we found four ColVLP-associated sequences that formed their unique cluster within the chromosomal homologs. An alignment and comparison of non-conserved sites revealed that these four ColVLP-associated *sitABCD* sequences were hybrids between a ColVLP and chromosomal homolog, where the initial section (*sitABC* and start of *sitD*) shared high conservation with the reference chromosomal sequence, and the latter section of *sitD* shared high conservation with the reference ColVLP-associated sequence (Figure 4G).

Taken together, our results indicate that while most virulence loci carried on the ColVLP are distinct from their chromosomal counterparts, rare homologous recombination events can occur, resulting in switching of the entire operon (as evidenced by *iroBCDEN*) or even a partial section of a virulence gene (as evidenced by *sitD*) between ColVLPs and the *E. coli* chromosome.'

We have also **modified Figure 4** by noting recombinant sequences and adding a panel to visualise the nucleotide identity of the non-conserved sites of the hybrid *sitABCD* genes compared to reference ColVLP- and chromosomal-associated sequences. **New text** added to the Figure Legend is underlined.

Figure 4. Phylogeny of the virulence genes (A) *iroBCDEN*, (B) *iucABCD/iutA*, (C) *shiF*, (D) *sitABCD*, (E) *iss*, and (F) *ompT*. Nucleotide sequences were extracted from the NCBI RefSeq (1,378 *E. coli* chromosomes) and 233 ColVLP collection, aligned using Clustal Omega, and visualised as a midpoint-rooted tree using FastTree and iTOL. Potential recombinant homologs are highlighted in blue. **(G) Nucleotide identity (%) comparing the non-conserved sites of the recombinant *sitABCD* operons**

with representative chromosomal and ColVLP homologs. Alignments between the four potential recombinant *sitABCD* operons (from NZ_CP019018.1, CP056868.1, NZ_CP032068.1, NZ_CP059908.1) and the representative homologs (ColVLP - NZ_CP026854.1, Chromosome - CP033884.1) were performed using CLC Main Workbench.

In addition, we have also **modified our discussion** to integrate these new findings:

Previous:

An intriguing aspect of our analysis was the distinct phylogenetic clustering of chromosomal- and ColVLP-encoded virulence genes, suggesting a lack of recombination between these regions (Figure 4). A limitation of this analysis was that the collection of chromosomal homologs was taken from completely sequenced *E. coli* genomes irrespective of ColVLP carriage and thus is likely predominantly from strains that do not carry ColVLPs. Indeed, a large collection of completely sequenced genomes belonging to ColVLP+ STs is lacking, making an assessment of recombination within these STs difficult. Thus, the full extent of recombination between chromosomal- and ColVLP-encoded virulence genes within these STs remains to be elucidated.

Changed:

An intriguing aspect of our analyses was the observation that some ColVLP-encoded virulence loci have recombined with their host chromosome, resulting in full operon switching (*iroBCDEN*; Figure 4A) to hybrid sequences (*sitABCD*; Figure 4G). Here, our database consisted of completely sequenced *E. coli* genomes irrespective of ColVLP carriage and is likely predominantly from strains that do not normally carry ColVLPs. As recombination between the ColVLP and chromosome can only occur when a strain carries a ColVLP, we suspect these recombination events occur at a rate higher than what our results suggest. To fully elucidate the full extent of recombination, a large collection of completely sequenced genomes belonging to ColVLP+ STs is required but currently lacking.

Specific comments:

Throughout: Incompatibility (Inc) is a phenotype. Sequence-based approaches cannot predict complex compatibility reactions, so use of the term is best avoided. On line 254, it would be more accurate to say that replicon types are being investigated, not Inc groups.

We thank the reviewer for pointing this out and have amended the manuscript accordingly (**changes underlined**):

“To investigate the replicons associated with ColVLP carriage, we plotted all significant co-carriage events in the 100ST database as a network. The top four replicons associated with ColVLP carriage were IncI1, IncX, IncB/O/K/Z, and IncH (Figure 3C). These co-carriage associations were also ST-dependent, as highlighted by the ColVLP-replicon associations of ST95 and ST131 (Clade B), two pandemic STs typically associated with ColVLP carriage, which were quantified with ST-specific databases of 2,118 and 344 draft genomes, respectively. In ColVLP+ ST95 (1268/2118; Figure 3F), the top four replicons associated with ColVLP carriage were (in order of magnitude): IncB/O/K/Z, IncX, IncH, and IncI1 (Figure 3D). In ColVLP+ ST131 Clade B genomes (191/344; Figure 3F), the top four replicons were: IncI1, IncH, IncX, and IncN (Figure 3E). Notably, strains of ST131 Clade B had higher replicon diversity compared to ST95. ST131 Clade B strains, in addition to carrying ColVLPs alone (31%), frequently contained combinations of ColVLP + IncI1 (24%); + IncI1 & IncH (10%); or + IncX (7%) (Figure 3I). In contrast, the majority of ST95 carried ColVLP alone (75%) or with an additional IncB/O/K/Z plasmid (13%) (Figure 3H). Taken together, the data demonstrates that ColVLP+ *E. coli*

strains frequently carry a diverse array of plasmid replicons, in part reflecting the existence of multiple co-resident plasmids in ST-specific patterns.”

Lines 73 and 138-141: Reference 16 from 2006 (PMID 16885466) is not very recent. Differences between ColV and ColBM plasmids have been known about for almost 20 years. Independent clustering of ColV and ColBM lineages is to be expected based on the existing literature. Considering that, should they be analysed together like this, or separately for greater resolution?

Our decision to analyse these plasmids together was based on the following: (i) ColBM plasmids still carry most of the virulence loci that define ColV plasmids, and thus have similar virulence characteristics; (ii) We could not be certain if a ColBM-negative plasmid had acquired ColBM and subsequently lost it, or if it never acquired ColBM in the first place.

Lines 86-87: Backbone vs acquired AMR determinants

Acquired antibiotic resistance genes are not part of plasmid backbones. The convergence of virulence and resistance in these plasmids is explained by the insertion of mobile genetic elements and their associated resistance genes INTO the ColV backbone. Suggest modifying this sentence to reflect this, as stating that these determinants “exist on” the backbone is inaccurate and will lead to confusion.

We have modified the sentence to reflect the comment as below (see **underlined changes**):

“Most importantly, several studies have reported increased antimicrobial resistance (AMR) among *E. coli* strains carrying ColVLPs (9, 25), with some studies highlighting the mobile genetic element-mediated insertion of these AMR determinants on the ColVLP plasmid backbone – a concerning convergence of virulence and resistance on a conjugative mobile genetic element (26).”

Line 115: Recognising FII replicons

The replicons called FIC(FII) here are FII. FIC is a misleading entry in the PlasmidFinder database. Compare the structures of these to reference FII replicons and see prior studies on ColV plasmids that call them FII-18 or FII-24. The FAB formula should be presented in the format FII:FIA:FIB.

We thank the reviewer for bringing this to our attention. We have confirmed that FIC-4 is detecting the same gene as FII-18, and have removed FIC replicons from Figure 1, Supplementary Figure 3 and Supplementary Table S4. We have also changed the FAB formula to be presented in the appropriate format.

Line 119-120: Assuming conjugative ability

Conjugative ability should not be assumed from the presence of transfer genes. Even when all genes are present, insertions or polymorphisms in the transfer region might impact a plasmid’s ability to transfer. See PMID 26855083 for an example in a ColV plasmid.

We agree, and we have amended the manuscript to include an appropriate level of caution (**changes are underlined**).

Figure 1 legend:

“Potential conjugative ability was predicted based on the presence of all genes required for F-plasmid conjugation (33) determined using BLASTn at a 90% nucleotide sequence identity and alignment length threshold, noting that a complete assessment of conjugation ability would require experimental validation.”

Results description:

More than half of the ColVLPs (52%) carried all genes required for F plasmid conjugation and are potentially conjugative (Figure 1), although transfer region polymorphisms and/or insertions may still impact transfer capability (33).

Lines 458-451: Conclusions

This seems like too strong a statement to make here. The convergence of virulence and antibiotic resistance in the context of ColV plasmids has been demonstrated and detailed in previous studies and there do not seem to be any mechanistic explanations presented here.

We have modified the conclusions statement as below (**new changes underlined**)

Previous:

“By demonstrating the carriage of multiple AMR genes on these plasmids, including co-integrate ColVLPs that harbour multiple replicons, our study describes a mechanism to explain plasmid-driven convergence of virulence and resistance in ExPEC.”

Changed:

“By demonstrating the carriage of multiple AMR genes on these plasmids, including co-integrate ColVLPs that harbour multiple replicons, our findings expand the role of ColVLPs in the plasmid-driven convergence of virulence and resistance in ExPEC.”